# BeyondMix: Leveraging Structural Priors and Long-Range Dependencies for Domain-Invariant LiDAR Segmentation

**Yujia Chen**[1,2], **Rui Sun**[3], **Wangkai Li**[1,2], **Huayu Mai**[1,2], **Si Chen**[1,2], **Zhuoyuan Li**[1,2],
**Zhixin Cheng**[1,2], **Tianzhu Zhang**[1,2*]
[1]University of Science and Technology of China
[2]National Key Laboratoray of Deep Space Exploration, Deep Space Exploration Laboratory
[3]Shenzhen International Graduate School, Tsinghua University
{yujia_chen, issunrui, lwklwk, mai556, sa23010094,
chengzhixin}@mail.ustc.edu.cn, tzzhang@ustc.edu.cn

## Abstract

Domain adaptation for LiDAR semantic segmentation remains challenging due to the complex structural properties of point cloud data. While mix-based paradigms have shown promise, they often fail to fully leverage the rich structural priors inherent in 3D LiDAR point clouds. In this paper, we identify three critical yet underexploited structural priors: permutation invariance, local consistency, and geometric consistency. We introduce BeyondMix, a novel framework that harnesses the capabilities of State Space Models (specifically Mamba) to construct and exploit these structural priors while modeling long-range dependencies that transcend the limited receptive fields of conventional voxel-based approaches. By employing space-filling curves to impose sequential ordering on point cloud data and implementing strategic spatial partitioning schemes, BeyondMix effectively captures domain-invariant representations. Extensive experiments on challenging LiDAR semantic segmentation benchmarks demonstrate that our approach consistently outperforms existing state-of-the-art methods, establishing a new paradigm for unsupervised domain adaptation in 3D point cloud understanding.

## 1 Introduction

LiDAR sensors maintain operational integrity under adverse conditions [41, 64, 6, 20] where camera-based perception fails. Semantic segmentation enables critical scene understanding for autonomous navigation safety [46, 39, 7]. Despite significant advancements through deep learning methodologies [16, 33, 79, 62, 1], LiDAR segmentation requires extensive annotated datasets—a substantial challenge given the prohibitive resource requirements for manually labeling point clouds comprising about $10^5$ points per scan. While synthetic data provides readily available annotations, it introduces domain shift, violating the i.i.d. assumption between training and deployment distributions in statistical learning theory [58], consequently degrading model performance. Unsupervised domain adaptation (UDA) techniques have been extensively studied to address this issue by transferring knowledge from a labeled source domain to an unlabeled target domain, and improve the model's performance on the target dataset without requiring additional annotations. The unstructured nature of LiDAR point clouds coupled with challenges in designing effective alignment methodologies renders UDA for LiDAR segmentation particularly difficult.

---

[*]Corresponding author

39th Conference on Neural Information Processing Systems (NeurIPS 2025).

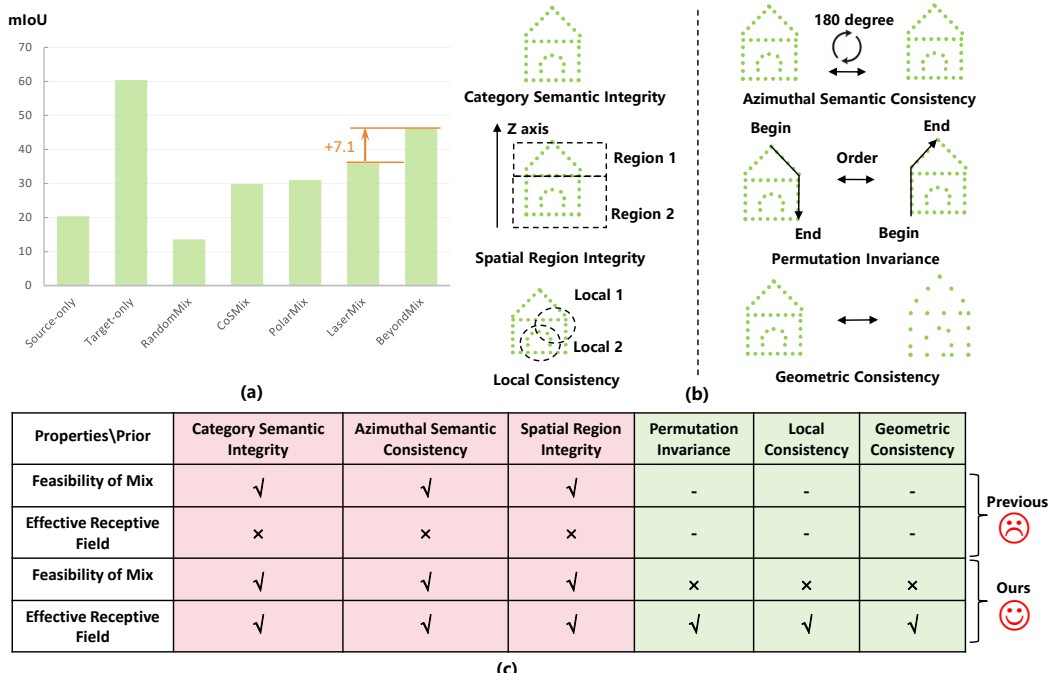

Figure 1: (a) Performance comparison across diverse mixing methodologies. (b) Schematic illustration of distinct structural priors. (c) Comparative analysis of properties between our proposed approach and existing methods.

In previous work, the Mix-based paradigm [48, 19, 68] has gained prominence due to algorithmic simplicity and empirical efficacy. These approaches generate intermediate domains by strategically combining source and target distributions, thereby smoothing decision boundaries and facilitating feature disentanglement [56]. Mix-based paradigm encompass various integration criteria: CoSMix [48] leverages semantic labels, PolarMix [68] utilizes azimuth angles, and LaserMix [19] incorporates inclination patterns for source-target domain fusion.

As illustrated in Figure 1 (a), which empirically validates the efficacy of structured Mix-based paradigms, we observe that naive random mixing of source-target domains results in substantial performance degradation. This deterioration occurs because the destruction of intrinsic geometric structure prior fundamentally compromises representational learning, rendering models incapable of extracting meaningful discriminative features. For example, when automobile and architectural elements are randomly mixed, their intersecting regions confound class-specific feature learning processes. We find that existing approaches leverage different structural priors: (1) category semantic integrity (CoSMix [48]), (2) azimuthal semantic consistency(PolarMix [68]), and (3) spatial regioin ntegrity(LaserMix [19]). However, the inherent structural priors in 3D LiDAR point clouds extend significantly beyond these three isolated dimensions, presenting substantial unexplored potential.

Through theoretical analysis, we uncover that previous paradigm have overlooked three critical structural priors, as illustrated in Figure 1 (b): (1) **Permutation Invariance Prior**, whereby point cloud representations should remain consistent regardless of acquisition trajectory or scanning order [42, 43] (e.g., different angular perspectives or sampling paths), preserving invariance to permutation operations. (2) **Local Consistency Prior**, whereby point cloud features should maintain consistency across different local spatial partitions [63, 23], independent of acquisition perspective or artificially defined spatial segmentation schemes; and (3) **Geometric Consistency Prior**, whereby LiDAR point cloud geometric structures (surface curvatures, normal vectors) should maintain stability under various processing operations [29, 71], with remaining points preserving critical geometric information even under partial masking. However, these three structural priors intuitively resist straightforward implementation through mix-based paradigm like previous work.

In addition, autonomous driving LiDAR datasets typically comprise hundreds of thousands of points per scan. The predominant methodology employs voxelization with U-Net architectures, wherein predictions for each voxel depend on its surrounding neighborhood—defined by the network's

receptive field—representing a careful balance between computational efficiency and performance. Our analysis demonstrates that Mix-based operations attain effectiveness by integrating scans from both the source and target domains, thereby ensuring that voxels receive informative signals from both domains within their receptive fields. When receptive fields encompass cross-domain contextual information, as illustrated in Figure 2, Point A, the model is constrained from relying solely on domain-specific cues, fundamentally promoting the learning of domain-invariant representations. We provide a theoretical justification in the Appendix. However, current mix-based methodologies suffer from a critical limitation: their sparse mixing patterns result in numerous voxels possessing receptive fields confined to single domains, significantly constraining the adaptation capacity of UDA frameworks. We conduct an exemplary analysis of LaserMix [19] which, despite achieving the most comprehensive mixing strategy and superior performance, still exhibits the aforementioned limitation. When LaserMix partitions scans into 2-6 regions based on inclination angle, each region spans approximately 360-1,300 voxels within the 4,000-voxels coordinate space. However, mainstream UNet architectures maintain receptive fields of merely 230 voxels, consequently resulting in numerous voxels being unable to simultaneously capture information from both domains. Similar observations have been documented across alternative mixing methodologies [48, 68]. As illustrated in Figure 2, Point B's receptive field fails to encompass both domains, fundamentally impeding the learning of domain-invariant representations.

Remarkably, we observe that recent advances in the State Space Model. SSMs [10] and Mamba [8] have emerged as promising architectural paradigms for sequence modeling, owing to their robust long-range modeling capabilities, and linear-time complexity. The Mamba architecture necessitates the transformation of unordered 3D LiDAR data into structured point sequences while preserving spatial proximity relationships. Our investigation reveals a twofold advantage: (1) Mamba's diverse scanning methodologies can potentially address the limitations inherent in mix-based approaches by constructing varied structural priors and effectively leveraging these priors. (2) Its sequential processing paradigm coupled with linear time complexity for modeling long-range dependencies enables comprehensive exploration of structural priors, thereby significantly expanding the effective receptive field, as illustrated by Point C in Figure 2.

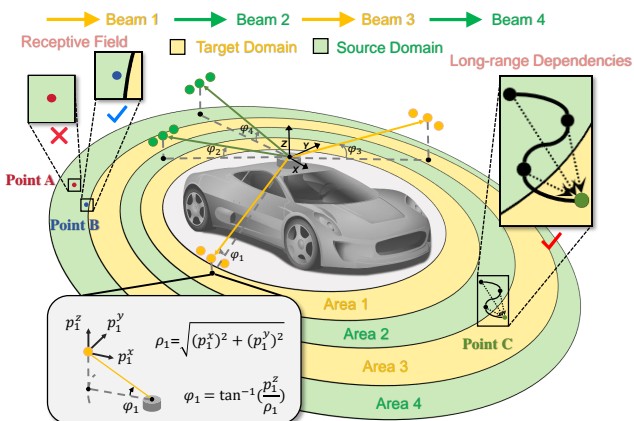

Figure 2: Comparative Analysis of Receptive Fields Across Different Methodologies. Point A illustrates a receptive field containing only a single domain, while Point B shows a receptive field spanning both domains. Point C demonstrates the implementation of SSMs, which effectively leverage long-range dependencies between points, resulting in an expansive receptive field.

In this work, we introduce **BeyondMix**, which leverages Mamba to construct structural priors **beyond** the capabilities of existing **mix**-based methods while simultaneously exploiting long-range dependency modeling to transcend limited voxel receptive fields for enhanced domain-invariant representation learning—as illustrated in Figure 1 (c). Specifically: Space-filling curves provide an established mechanism for imposing sequential ordering on point cloud data by mapping multidimensional space to one-dimensional sequences while maintaining locality properties. Two predominant approaches—Hilbert curves [13] and Z-order curves [34]—serve distinct computational purposes: Hilbert curves optimize range queries, while Z-order curves facilitate hierarchical indexing (visualization analysis in Appendix). (1) Diverse space-filling curves induce different spatial proximity relationships that inherently satisfy **permutation invariance prior** requirements. (2) Leveraging the local semantic similarity property of LiDAR point clouds [19], we implement both cylindrical partitioning (via inclination angles) and rectangular partitioning (via Z-axis coordinates) for each scan. These different sub-region divisions enable the construction of **local consistency prior**. (3) Recognizing that scans with varying densities maintain invariant structural properties, we formulate **geometric consistency prior**.

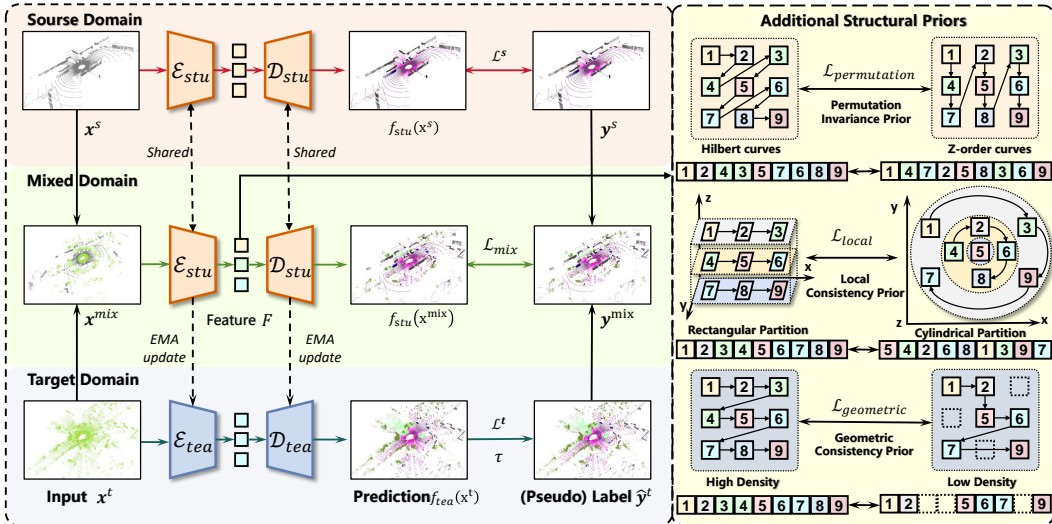

Figure 3: BeyondMix overview. For labeled source domain data $x^s$ and mixed domain data $x^{mix}$, we compute supervised loss $\mathcal{L}^s$ and $\mathcal{L}_{\mathrm{mix}}$ through the student network $f_{stu}$, while for unlabeled target data $x^t$, we compute adaptive loss $\mathcal{L}^t$ via both student $f_{stu}$ and teacher $f_{tea}$ networks. Additionally, we design three additional structural priors: (1) Permutation Invariance Prior across different Lidar point orderings, (2) Local Consistency Prior between various structural partitioning schemes, and (3) Geometric Consistency Prior across different point density distributions.

Our contributions can be summarized as: (1) **Problem Identification** (WHY): We provide a dual-perspective analysis of existing mix-based methods in UDA for LiDAR point cloud segmentation, revealing insufficient exploitation and utilization of structural priors; (2) **Methodology Design** (WHAT): We propose BeyondMix, leveraging Mamba to construct structural priors beyond mix-based methods while thoroughly exploiting long-range dependencies to overcome limited voxel receptive fields; (3) **Implementation** (HOW): Extensive experiments demonstrate that BeyondMix consistently achieves SOTA performance on two challenging LiDAR semantic segmentation benchmarks.

## 2 Related Work

**UDA for Point Cloud Segmentation**. UDA for point cloud semantic segmentation follows two main paradigms: (1) Range-view methods that address domain shift in 2D projections, evolving from Cycle-GAN techniques (ePointDA [77]) and geodesic alignment (SqueezeSegV2 [65]) to domain-specific knowledge extraction (Gated [45], CCL [22]); and (2) 3D-based methods operating directly on point clouds through density normalization (C&L [70], PCT [69]), statistical alignment (DGT [73]), and domain-invariant feature learning [59, 61, 72, 73, 78, 68, 19, 48]. This latter category encompasses three key approaches: adversarial training methodologies (ADVENT [59], FADA [61], MRNet [78]) using discriminator networks to align feature representations; prototype guidance techniques (PMAN [72], PCAN [73]) that extract source domain prototypes to enhance target domain learning; and mixup strategies (CoSMix [48], PolarMix [68], LaserMix [19]) that integrate domains through various criteria. While mix-based methods effectively construct intermediate domains and partially leverage structural priors, their efficacy remains fundamentally constrained by insufficient exploration and utilization of the complete prior space available in LiDAR point cloud representations. More related work is provided in the Appendix.

## 3 Method

In this section, we first formalize the standard self-training paradigm for 3D UDA semantic segmentation and Mamba, respectively (Sec 3.1). Then, we introduce our new 3D LiDAR UDA framework, BeyondMix, a novel approach that thoroughly exploits and leverages structural priors unattainable by previous mix-based methods, while alleviating the limited receptive field issues inherent in prior approaches (Sec 3.2). Finally, we provide comprehensive details regarding training and inference procedures (Sec 3.2). The overall framework of our proposed approach is illustrated in Figure 3.

### 3.1 Preliminaries

**Self-Training (ST) for 3D UDA semantic segmentation.** Due to its empirical efficacy and convergence stability, the self-training paradigm has emerged as the predominant framework for 3D unsupervised domain adaptation tasks. For a labeled source LiDAR point cloud scan $x^s$, we employ the standard cross-entropy loss function to derive precise semantic segmentation predictions for each individual voxel:

$$\mathcal{L}^s = -\frac{1}{N} \sum_{i=1}^{N} \sum_{k=0}^{K} y_i^s \log P_{i,k}^s, \tag{1}$$

where $K$ denotes the total number of semantic classes, $y_i^s$ represents the ground-truth semantic label, and $P_{i,k}^s$ represents the class-conditional probability prediction for the $i$-th voxel across the $K$ semantic categories.

For an unlabeled target domain scan $\mathbf{x}^t$, we leverage pseudo-labeling as a supervised learning strategy to facilitate domain adaptation:

$$\mathcal{L}^t = -\frac{1}{N} \sum_{i=1}^{N} \sum_{k=0}^{K} \hat{y}_i^t \log P_{i,k}^t. \tag{2}$$

The pseudo-labels are generated by the teacher network $f_{tea}$, yielding the network's class-conditional probability output $P_t(\mathbf{x}_i^t)$. We then apply a confidence thresholding mechanism with a predefined threshold $\tau$ to filter these pseudo-labels, selecting only the most reliable pseudo-annotations:

$$\hat{y}_i = \begin{cases} \arg\max\limits_{c} P_t^{(c)}(\mathbf{x}_i^t), & \text{if } \max\limits_{c} P_t^{(c)}(\mathbf{x}_i^t) \geq \tau \\ 0, & \text{otherwise} \end{cases} \tag{3}$$

where $\tau \in [0, 1]$ is a confidence threshold. The teacher network's weights are dynamically updated through an Exponential Moving Average (EMA) strategy implemented by the student network $f_{stu}$, ensuring smooth and stable knowledge transfer: $\theta_i^{tea} = \alpha\theta_{i-val}^{tea} + (1-\alpha)\theta_i^{stu}$, where $\alpha \in [0,1)$ is the momentum coefficient, and $\theta$ represents the network parameters, $i$ denotes the current training iteration, and $val$ represents the update interval.

**Mamba.** The SSMs is initially introduced in the field of control engineering to model dynamic systems. Specifically, the SSMs in deep learning encompass three key variables: the input $x(t)$, the latent state representation $h(t)$, and the output $y(t)$. The system can be defined as follows:

$$h'(t) = \mathbf{A}h(t) + \mathbf{B}x(t), \quad y(t) = \mathbf{C}h'(t). \tag{4}$$

where $\mathbf{A}$ represents the state transition matrix that describes how the system states volve, $\mathbf{B}$ denotes the control matrix that describes the influence of the inputs on the states, and $\mathbf{C}$ defines state impact on outputs. To handle discrete-time sequence data inputs, the Zero-Order Hold is typically used:

$$h_t = \overline{\mathbf{A}}h_{t-1} + \overline{\mathbf{B}}x_t, y_t = \overline{\mathbf{C}}h_t, \quad \overline{\mathbf{A}} = \exp(\boldsymbol{\Delta}\mathbf{A}), \overline{\mathbf{B}} = (\boldsymbol{\Delta}\mathbf{A})^{-1}(\exp(\boldsymbol{\Delta}\mathbf{A}) - \mathbf{I})\boldsymbol{\Delta}\mathbf{B}, \tag{5}$$

where $\boldsymbol{\Delta}$ represents the temporal discretization interval. However, due to the linear time invariant nature of SSM, the parameters ($\boldsymbol{\Delta}$, $\mathbf{A}$, $\mathbf{B}$, $\mathbf{C}$) remain fixed across all time steps, which limits the expressive capacity of SSM. To overcome this limitation, Mamba [8] introduces a hardware-aware scan algorithm to achieve near-linear complexity and a selection mechanism that treats the parameters ($\boldsymbol{\Delta}$, $\mathbf{B}$, $\mathbf{C}$) as functions of the input, effectively transforming the SSM into a time-varying model:

$$h_t = \phi_{\overline{\mathbf{A}}}(x_t)h_{t-1} + \phi_{\overline{\mathbf{B}}}(x_t)x_t, y_t = \phi_{\mathbf{C}}(x_t)h_t, \tag{6}$$

where $\phi_{\overline{\mathbf{A}}}(x_t), \phi_{\overline{\mathbf{B}}}(x_t)$ and $\phi_{\mathbf{C}}(x_t)$ denote the parameter matrices are dependent on the inputs $x_t$.

### 3.2 BeyondMix

BeyondMix selects the most prevalent mixing strategy LaserMix [19] to generate mixed scans and labels by combining source and target domain data, thereby constructing mix-based structural priors:

$$x^{mix} = \text{LaserMix}(x^s, x^t), \quad y^{mix} = \text{LaserMix}(y^s, y^t), \quad \mathcal{L}^{mix} = -\frac{1}{N} \sum_{i=1}^{N} \sum_{k=0}^{K} y_i^{mix} \log P_{i,k}^{mix}, \tag{7}$$

where $\mathcal{L}_{\text{mix}}$ is the cross-entropy supervision error for the mixed scan. The discriminative feature representation $F \in \mathbb{R}^{M \times C}$ (whose spatial proximity relationships still preserve the 3D structure) is then obtained by processing the mixed scan $x^{mix}$ through the encoder network, where $M$ is the number of features and $C$ is the feature dimension. To further enhance the learning of domain-invariant

representations in discriminative features, we propose three key structural priors beyond conventional mixing approaches—permutation invariance, local consistency, and geometric consistency—while simultaneously modeling long-range dependencies through efficient sequence processing mechanisms.

**Permutation Invariance Prior.** Diverse space-filling curves, due to their different computational approaches to spatial proximity, induce different 1D sequential arrangements that inherently satisfy permutation invariance prior requirements [42, 43]. Simultaneously, their sequential ordering thoroughly leverages structural prior cues and explores long-range dependencies. Specifically, let $\pi_H : \{1, \ldots, M\} \to \{1, \ldots, M\}$ be the Hilbert curve permutation [13] and $\pi_Z : \{1, \ldots, M\} \to \{1, \ldots, M\}$ be the Z-order permutation [34]. By applying the Hilbert and Z-order permutations $\pi_H$ and $\pi_Z$, we obtain the reordered features $F_H = F[\pi_H]$ and $F_Z = F[\pi_Z]$, respectively. For each ordered sequence, the Mamba processes the features sequentially to generate permutation-sensitive features:

$$O_H = \mathcal{M}(F_H) \in \mathbb{R}^{M \times C}, \quad O_Z = \mathcal{M}(F_Z) \in \mathbb{R}^{M \times C}, \tag{8}$$

where $\mathcal{M}(\cdot)$ represents the Mamba model. To enforce consistency across differentially ordered input features, we employ inverse mappings of the original permutations, thereby reconstructing representations in the default ordering for equivariance constraints: $\tilde{O}_H = O_H[\pi_H^{-1}]$ and $\tilde{O}_Z = O_Z[\pi_Z^{-1}]$. $\mathcal{L}_{\text{permutation}}$ is then formulated as the mean squared error (MSE) between corresponding features across sequences:

$$\mathcal{L}_{\text{permutation}} = \left\| \tilde{O}_H - \tilde{O}_Z \right\|_2^2 \tag{9}$$

**Local Consistency Prior.** The spatial distribution of objects and backgrounds in real-world LiDAR scans exhibits pronounced spatial correlations and semantic regularities [63, 23, 19]. Specifically, the semantic composition of LiDAR point clouds demonstrates strong locality-based patterns, wherein objects and backgrounds within proximal spatial regions tend to share characteristic semantic attributes. For instance, close-range areas of the LiDAR scan predominantly manifest road-like semantics, while distant regions typically encompass more complex urban landscape elements such as buildings and vegetation. Moreover, near-ground spatial zones predominantly feature road and sidewalk semantics, with minimal representation of elevated structures like buildings. This indicates that 3D LiDAR point clouds possess distinct height and range attribute priors, which encourages us to leverage structural priors for partitioning point cloud scans. However, regardless of the partitioning approach, LiDAR point cloud features should maintain consistency. Therefore, we construct local consistency priors and adopt two partitioning strategies: first, using the Z-axis to leverage category height attributes for division; second, using elevation angle to partition based on category-to-sensor distance attributes.

First, we employ elevation angle $\theta$ and Z-axis height $z$ as primary stratification mechanisms for segmenting the discriminative feature representation $F \in \mathbb{R}^{M \times C}$ into distinct cylindrical and rectangular regions. We accomplish this by quantizing $\theta_i$ and $z_i$ into disjoint intervals. The cylindrical partitioning is defined as: $\mathcal{R}_\theta^l = \left\{ F_i \mid \theta_{\min}^l \leq \theta_i < \theta_{\max}^l \right\}, l = 1, \ldots, L$, where $\theta_{\min}^l$ and $\theta_{\max}^l$ define the angular bounds of the $l$-th region. Similarly, the rectangular partitioning is formulated as: $\mathcal{R}_z^v = \left\{ F_i \mid z_{\min}^v \leq z_i < z_{\max}^v \right\}, v = 1, \ldots, V$, where $z_{\min}^v$ and $z_{\max}^v$ define the height bounds of the $v$-th region. Subsequently, we implement Region-wise Hilbert Ordering within each segmented region, employing a Hilbert curve-based approach to systematically arrange features while preserving intra-region spatial proximity and geometric relationships. We then concatenate these ordered subregions according to their regional sequence to generate distinct permutation schemes $\pi_L$ and $\pi_V$ that encode different structural priors. Applying these permutations yields reordered feature representations $F_L = F[\pi_L]$ and $F_V = F[\pi_V]$. Following Equation 8, we generate permutation-sensitive features $O_L$ and $O_V$. To establish local consistency prior, we map these features back to the default ordering through inverse permutation: $\tilde{O}_L = O_H[\pi_L^{-1}]$ and $\tilde{O}_V = O_Z[\pi_V^{-1}]$. The local consistency prior loss is formulated as follows:

$$\mathcal{L}_{\text{local}} = \left\| \tilde{O}_L - \tilde{O}_V \right\|_2^2 \tag{10}$$

**Geometric Consistency Prior.** Despite their sparsity and lack of ordering, 3D LiDAR point clouds exhibit strong geometric redundancy due to underlying surface continuity. Point clouds typically form coherent structures with local smoothness and global consistency, enabling reliable inference of original geometry even from partially masked representations [42, 29, 71]. We leverage this property to establish geometric consistency priors through masking operations. We generate a density-perturbed version $\tilde{F}$ by applying a random binary mask $Mask \in \{0, 1\}^M$:

$$\tilde{F} = \{F_i \mid Mask_i = 1\}, \quad \text{where } Mask_i \sim \text{Bernoulli}(1 - \rho), \tag{11}$$

Table 1: Comparison results of SynLiDAR → SemanticKITTI adaptation in terms of mIoU. The highest scores for each semantic class are highlighted using a color-coding system: Source only results are marked in yellow, Target only results in gray, and the overall best performance metrics in blue.

| Methods | mIoU | car | bi.cle | mt.cle | truck | oth-v. | pers. | bi.clst | mt.clst | road | parki. | sidew. | other-g. | build. | fence | veget. | trunk | terr. | pole | traf. |
|---|---|---|---|---|---|---|---|---|---|---|---|---|---|---|---|---|---|---|---|---|
| Source only | 20.4 | 35.9 | 7.5 | 10.7 | 0.6 | 2.9 | 13.3 | 44.7 | 3.4 | 21.8 | 6.9 | 29.6 | 0.0 | 34.1 | 7.4 | 62.9 | 26.0 | 35.5 | 30.3 | 14.1 |
| Target only | 60.4 | 96.2 | 17.6 | 55.9 | 79.5 | 51.4 | 65.5 | 84.9 | 2.8 | 93.2 | 38.5 | 79.8 | 1.9 | 90.4 | 57.3 | 86.8 | 65.4 | 72.7 | 64.3 | 43.2 |
| AdaptSegNet [57] | 27.9 | 52.1 | 10.8 | 11.2 | 2.6 | 9.6 | 15.1 | 35.9 | 2.6 | 62.2 | 10.4 | 41.3 | 0.1 | 58.1 | 17.1 | 68.0 | 38.4 | 38.7 | 35.9 | 20.4 |
| CLAN [32] | 30.5 | 51.0 | 15.8 | 16.8 | 2.2 | 7.8 | 18.7 | 46.8 | 3.0 | 68.9 | 11.1 | 44.9 | 0.1 | 59.6 | 17.5 | 71.7 | 41.1 | 44.0 | 37.7 | 19.8 |
| ADVENT [59] | 30.5 | 59.9 | 13.8 | 14.6 | 3.0 | 8.0 | 17.7 | 45.8 | 3.0 | 67.6 | 11.3 | 45.6 | 0.1 | 61.7 | 15.8 | 72.4 | 41.5 | 47.0 | 34.5 | 15.3 |
| FADA [61] | 25.6 | 49.9 | 6.7 | 5.1 | 2.5 | 10.0 | 5.7 | 26.6 | 2.3 | 65.8 | 10.8 | 37.8 | 0.1 | 60.3 | 21.5 | 60.4 | 37.2 | 31.9 | 35.4 | 17.4 |
| MRNet [78] | 28.3 | 49.5 | 11.0 | 12.2 | 2.2 | 8.6 | 16.0 | 46.4 | 2.7 | 60.0 | 10.5 | 41.9 | 0.1 | 55.1 | 16.5 | 68.1 | 38.0 | 40.7 | 36.5 | 20.8 |
| PMAN [72] | 33.7 | 71.0 | 14.9 | 24.8 | 1.6 | 6.6 | 23.6 | 61.1 | 5.5 | 75.3 | 10.5 | 54.1 | 0.1 | 47.9 | 17.4 | 69.6 | 38.6 | 61.5 | 37.0 | 18.6 |
| CoSMix [48] | 29.9 | 56.4 | 10.2 | 20.8 | 2.1 | 13.0 | 25.6 | 41.3 | 2.2 | 67.4 | 8.2 | 43.4 | 0.0 | 57.9 | 12.2 | 68.4 | 44.8 | 35.0 | 42.1 | 17.0 |
| PolarMix [68] | 31.0 | 76.3 | 8.4 | 17.8 | 3.9 | 6.0 | 26.6 | 40.8 | 15.9 | 70.3 | 0.0 | 44.4 | 0.0 | 68.4 | 14.7 | 69.6 | 38.1 | 37.1 | 40.6 | 10.6 |
| LaserMix [19] | 36.0 | 90.3 | 7.8 | 37.2 | 2.3 | 2.4 | 40.6 | 49.1 | 5.1 | 80.5 | 9.9 | 57.4 | 0.0 | 57.6 | 3.4 | 77.6 | 46.6 | 60.1 | 42.0 | 13.6 |
| PCAN [73] | 37.0 | 85.0 | 17.5 | 27.4 | 10.4 | 11.9 | 27.5 | 63.7 | 2.6 | 78.1 | 13.5 | 50.1 | 0.1 | 68.5 | 20.0 | 76.2 | 41.3 | 45.7 | 41.0 | 21.8 |
| DGT-ST [73] | 43.1 | 92.9 | 17.3 | 43.4 | 15.0 | 6.1 | 49.2 | 54.2 | 4.2 | 86.4 | 19.1 | 62.3 | 0.0 | 78.2 | 9.2 | 83.3 | 56.0 | 59.1 | 51.2 | 32.2 |
| **BeyondMix (Ours)** | 45.4 | 93.3 | 17.1 | 44.9 | 16.1 | 14.7 | 51.0 | 61.4 | 8.3 | 87.0 | 21.3 | 68.2 | 0.0 | 78.1 | 17.1 | 83.1 | 59.5 | 59.8 | 52.6 | 30.3 |
| **BeyondMix++ (Ours)** | 46.2 | 94.0 | 15.1 | 47.2 | 17.6 | 16.5 | 55.2 | 59.9 | 6.8 | 87.0 | 24.1 | 69.3 | 0.0 | 79.3 | 14.7 | 81.8 | 61.0 | 62.8 | 53.3 | 32.1 |
| Δ ↑ | +23.8 | +58.1 | +7.6 | +36.5 | +17.0 | +13.6 | +41.9 | +15.2 | +3.4 | +65.2 | +14.2 | +39.7 | +0.0 | +45.2 | +7.3 | +18.9 | +35.0 | +27.3 | +23.0 | +18.0 |

and $\rho \in [0, 1)$ is the subsampling rate. We then apply Hilbert curve ordering to generate density-sensitive feature representations $O$ and $\tilde{O}$:

$$O = \mathcal{M}(F) \in \mathbb{R}^{M \times C}, \quad \tilde{O} = \mathcal{M}(\tilde{F}) \in \mathbb{R}^{M' \times C}, \quad \text{where } M' = \sum_{i=1}^{M} Mask_i. \tag{12}$$

For the computation of geometric consistency loss, we consider only the unmasked portions of the representation, thereby ensuring that the loss function evaluates only regions with reliable geometric information. Let $\mathcal{I} = \{i \mid Mask_i = 1\}$ be the index set of unmasked points. We define the masked subset of $O$ as $O_{\mathcal{I}} = \{O_i \mid i \in \mathcal{I}\}$. The loss is computed as:

$$\mathcal{L}_{\text{geometric}} = \frac{1}{M'} \sum_{i=1}^{M'} \left\| O_{\mathcal{I}}^{(i)} - \tilde{O}^{(i)} \right\|_2^2. \tag{13}$$

**Training and Inference.** The total loss for our training is given by:

$$\mathcal{L}_{\text{total}} = \mathcal{L}^s + \mathcal{L}^t + \lambda_{mix}\mathcal{L}_{\text{mix}} + \lambda_{prior}(\mathcal{L}_{\text{permutation}} + \mathcal{L}_{\text{local}} + \mathcal{L}_{\text{geometric}}), \tag{14}$$

where $\lambda_{mix}$ and $\lambda_{prior}$ are the balancing coefficients for the loss. For inference, we only use the original network structure.

**Expansion to other Mix-based methods.** Existing mix-based methods each possess distinct structural priors. Thoroughly exploiting and leveraging a diverse range of structural priors can significantly enhance model performance. Based on this insight, we extend BeyondMix to BeyondMix++, which randomly selects from various existing mixing strategies when generating intermediate domains, including CoSMix [48], LaserMix [19], and PolarMix [68].

## 4 Experiments

### 4.1 Experimental Setup

**Datasets.** We utilize three common LiDAR datasets, performing two synthetic-to-real UDA tasks. (1) **SynLiDAR** [69] is a synthetic dataset containing 198,396 point clouds with 32 semantic classes across 13 sequences. It simulates a Velodyne HDL-64E LiDAR sensor. We follow the authors' instructions [69] and use 19,840 point clouds for training and 1,976 for validation. (2) **SemanticKITTI** [2] is a real-world dataset collected in Karlsruhe, Germany, featuring 28 semantic classes. It was captured using a Velodyne HDL-64E LiDAR sensor. We use sequences 00-10 (19,130 scans) for training, while sequence 08 (4,071 scans) serves as validation. (3) **SemanticPOSS** [37] consists of

Table 2: Comparison results of SynLiDAR → SemanticPOSS adaptation in terms of mIoU. The highest scores for each semantic class are highlighted using a color-coding system: Source only results are marked in yellow, Target only results in gray, and the overall best performance metrics in blue.

| Methods | mIoU | bi.clst | car | trunk | veget. | traf. | pole | garb. | build. | cone. | fence | bi.cle | ground | pers. |
|---|---|---|---|---|---|---|---|---|---|---|---|---|---|---|
| Source only | 38.3 | 47.2 | 43.6 | 37.8 | 70.3 | 11.1 | 33.8 | 19.5 | 67.9 | 11.2 | 19.9 | 9.6 | 77.9 | 47.8 |
| Target only | 57.3 | 61.6 | 75.1 | 48.4 | 77.9 | 47.7 | 37.8 | 29.9 | 77.8 | 37.7 | 51.2 | 54.9 | 81.2 | 63.9 |
| AdaptSegNet [57] | 39.3 | 43.9 | 48.2 | 39.0 | 69.6 | 15.5 | 33.6 | 21.3 | 64.3 | 12.7 | 25.0 | 11.6 | 76.0 | 49.9 |
| CLAN [32] | 39.5 | 43.9 | 46.6 | 41.3 | 71.0 | 15.1 | 34.3 | 20.4 | 69.6 | 9.5 | 23.2 | 12.0 | 75.1 | 51.3 |
| ADVENT [59] | 40.1 | 44.6 | 47.6 | 40.3 | 71.2 | 15.6 | 35.6 | 22.0 | 68.4 | 10.6 | 25.9 | 10.4 | 76.7 | 52.3 |
| FADA [61] | 37.6 | 39.6 | 41.2 | 38.8 | 69.2 | 16.3 | 32.1 | 18.1 | 67.9 | 11.5 | 22.0 | 13.0 | 71.4 | 47.9 |
| MRNet [78] | 39.4 | 43.5 | 47.2 | 39.1 | 70.4 | 15.5 | 32.8 | 22.0 | 66.1 | 13.2 | 24.2 | 11.2 | 76.8 | 50.0 |
| PMAN [72] | 46.5 | 52.6 | 61.5 | 46.8 | 75.1 | 18.8 | 36.5 | 21.4 | 74.7 | 18.3 | 25.8 | 37.5 | 73.7 | 61.9 |
| CoSMix [48] | 44.6 | 53.6 | 47.6 | 44.8 | 75.1 | 16.8 | 37.9 | 25.3 | 72.7 | 19.9 | 39.7 | 10.8 | 80.0 | 56.5 |
| PolarMix [68] | 32.6 | 39.1 | 25.0 | 11.9 | 64.2 | 5.8 | 29.6 | 15.3 | 44.8 | 13.3 | 23.8 | 10.7 | 79.0 | 30.4 |
| LaserMix [19] | 45.5 | 58.4 | 61.3 | 47.7 | 69.0 | 21.9 | 39.5 | 30.9 | 61.0 | 16.1 | 36.5 | 7.1 | 79.5 | 62.6 |
| PCAN [73] | 44.4 | 48.6 | 62.1 | 37.5 | 74.0 | 23.9 | 31.4 | 22.2 | 76.9 | 6.5 | 41.9 | 11.9 | 79.1 | 61.2 |
| DGT-ST [73] | 50.8 | 55.1 | 70.7 | 46.1 | 74.2 | 30.1 | 36.3 | 44.1 | 81.0 | 4.3 | 62.8 | 10.3 | 78.5 | 67.2 |
| **BeyondMix (Ours)** | 52.9 | 57.6 | 68.5 | 46.2 | 76.9 | 32.1 | 39.2 | 40.5 | 79.9 | 21.4 | 62.7 | 12.8 | 79.7 | 67.1 |
| **BeyondMix++ (Ours)** | 53.7 | 59.1 | 69.8 | 48.0 | 76.9 | 37.4 | 38.5 | 42.8 | 81.0 | 21.5 | 61.6 | 14.7 | 79.3 | 67.3 |
| Δ ↑ | +15.4 | +11.9 | +26.2 | +10.2 | +6.6 | +26.3 | +4.7 | +23.3 | +13.1 | +10.3 | +41.7 | +5.1 | +1.4 | 19.4 |

2,988 annotated real-world point clouds with 14 semantic classes, collected at Peking University, China, using a Pandora 40-line LiDAR sensor. We use sequence 03 for validation and all remaining sequences for training.

**Evaluation protocol.** Following virtual-to-real adaptation protocols established in previous works [69, 48, 73], we conduct two UDA experiments: SynLiDAR → SemanticPOSS (mapping to 14 segmentation classes) and SynLiDAR → SemanticKITTI (mapping to 19 segmentation classes). We adopt the widely accepted Intersection over Union (IoU) metric for semantic segmentation assessment. The IoU is calculated separately for each semantic category, providing class-specific insight into segmentation quality.

**Implementation.** Our implementation framework leverages PyTorch [38] and MinkowskiEngine [5], running on a single NVIDIA RTX A6000 GPU with 48GB VRAM. The semantic segmentation backbone employs a U-Net architecture MinkUNet34. For structural priors and Long-Range dependencies, we adhered to the default configuration with the Mamba architecture parameterized as follows: input dimension of 256, hidden state dimension of 256, convolutional kernel width of 4, and expansion factor of 2. In alignment with established practices in recent literature [48, 72, 73], we utilize raw $XYZ$ coordinates as the primary input features. Our processing pipeline maintains a consistent voxel size of 0.05m while accommodating variable point cloud densities without explicit size constraints. The optimization process employs the Adam optimizer [17] with a starting learning rate of 2.5e-4, governed by a polynomial decay schedule with power coefficient 0.9. Comprehensive experimental configuration details are available in the Appendix.

## 4.2 Comparative Study

We conduct comprehensive experiments to evaluate the performance of domain adaptation from SynLiDAR to SemanticKITTI and SemanticPOSS datasets. The mIoU and per-class IoU scores are reported in Tables 1 and 2. Qualitative Results can be found in the Appendix.

**Quantiative Results: (1) SynLiDAR → SemanticKITTI Adaptation.** Our proposed BeyondMix++ achieves state-of-the-art performance with an mIoU of 46.2%, surpassing all baseline methods by a significant margin (+23.8% over the source-only model). Notably, BeyondMix++ demonstrates exceptional improvements in challenging categories such as car (+58.1%), person (+41.9%), and road (+65.2%), highlighting its robustness in adapting to diverse object scales and geometries. Compared to the recent DGT-ST [73], BeyondMix++ attains consistent gains across 13 out of 19 classes. **(2) SynLiDAR → SemanticPOSS Adaptation.** BeyondMix++ achieves an mIoU of 53.7%, outperforming existing methods by up to +8.2% (vs. LaserMix[19]). Remarkably, our method exhibits superior generalization on sparse objects(*e.g.,* traffic-sign). Furthermore, BeyondMix++ significantly narrows the performance gap between source-only and target-only models, indicating its capability to achieve more effective domain-invariant representation learning.

## 4.3 Ablation Study

Without loss of generalizability, we stick with SynLiDAR $\rightarrow$ semanticKITTI in our ablations. More ablation studies are presented in the Appendix.

**The Effect of Components.** To validate the effectiveness of the proposed components in Beyond-Mix++, we conduct a systematic ablation study by progressively integrating different loss functions. Using the self-training-based DGT-ST [73] as our baseline, as shown in Table 3, we observe that employing the mixed method $\mathcal{L}_{\mathrm{mix}}$ yields a 1.0% improvement. Exploring and constructing different structural priors through $\mathcal{L}_{\mathrm{permutation}}$, $\mathcal{L}_{\mathrm{local}}$, and $\mathcal{L}_{\mathrm{geometric}}$ contributes improvements of 1.4%, 1.6%, and 1.3% respectively, indicating that each structural prior can help with domain-invariant representation learning. The pairwise combinations of these components yield stronger gains (1.9-2.7%), demonstrating synergistic effects in feature representation. Our full method ultimately achieves a substantial performance of 46.2% mIoU, representing a significant gain of 3.1% over the baseline. These

Table 3: Comparative study of different components.

| # | $\mathcal{L}_{\mathrm{mix}}$ | $\mathcal{L}_{\mathrm{permutation}}$ | $\mathcal{L}_{\mathrm{local}}$ | $\mathcal{L}_{\mathrm{geometric}}$ | mIoU(%) |
|---|---|---|---|---|---|
| (1) | | | | | 43.1 |
| (2) | ✓ | | | | 44.1 (+1.0) |
| (3) | ✓ | ✓ | | | 44.5 (+1.4) |
| (4) | ✓ | | ✓ | | 44.7 (+1.6) |
| (5) | ✓ | | | ✓ | 43.4 (+1.3) |
| (6) | ✓ | ✓ | ✓ | | 45.4 (+2.3) |
| (7) | ✓ | ✓ | | ✓ | 45.8 (+2.7) |
| (8) | ✓ | | ✓ | ✓ | 45.0 (+1.9) |
| (9) | ✓ | ✓ | ✓ | ✓ | **46.2 (+3.1)** |

results confirm the complementary nature of our proposed components in helping the model learn more comprehensive domain-invariant representations under different structural priors, through Permutation Invariance Prior, Local Consistency Prior, and Geometric Consistency Prior.

Table 4: Comparison between different cylindrical partitioning region number $L$.

| Num $L$ | mIoU(%) |
|---|---|
| 3 | 45.4 (-0.8) |
| 4 | 45.6 (-0.6) |
| 5 | **46.2** |
| 6 | 45.7(-0.5) |

Table 5: Comparison between different rectangular partitioning region number $M$.

| Num $M$ | mIoU(%) |
|---|---|
| 2 | 45.3 (-0.9) |
| 3 | **46.2** |
| 4 | 46.0 (-0.2) |
| 5 | 45.8 (-0.4) |

Table 6: Comparison between different Mask Ratios.

| Mask(%) | mIoU(%) |
|---|---|
| 40 | 45.9 (-0.3) |
| 50 | **46.2** |
| 60 | 45.5 (-0.8) |
| 70 | 44.9 (-1.3) |
| 80 | 44.3 (-1.9) |

Table 7: Comparison between different mix strategies.

| # | CosMix | PolarMix | LaserMix | mIoU(%) |
|---|---|---|---|---|
| (1) | ✓ | | | 45.1 (-1.1) |
| (2) | | ✓ | | 44.8 (-1.4) |
| (3) | | | ✓ | 45.4 (-0.8) |
| (4) | ✓ | ✓ | | 45.7 (-0.5) |
| (5) | ✓ | | ✓ | 46.0 (-0.2) |
| (6) | | ✓ | ✓ | 45.9 (-0.3) |
| (7) | ✓ | ✓ | ✓ | **46.2** |

**The Effect of Local Consistency Prior.** Tables 4 and 5 evaluate different partitioning configurations for our Local Consistency Prior. The optimal setup combines 5 cylindrical regions and 3 rectangular regions, achieving an ideal balance between segregating semantically distinct areas while preserving contextual information within regions. This configuration provides the best partitioning of spatial correlations and semantic regularities, helping the model improve UDA performance.

**The Effect of Geometric Consistency Prior.** Table 6 reveals that a 50% mask ratio achieves optimal performance (46.2% mIoU) for density consistency regularization. A lower ratio (40%) shows minor degradation (-0.3%) due to insufficient density variation, while higher ratios (60-80%) cause increasingly significant performance drops (-0.8% to -1.9%) as excessive point removal compromises structural information. The 50% mask ratio effectively balances creating meaningful density variation while preserving sufficient structural context for learning density-invariant representations across different geometric structure prior.

**The Effect of Mix Strategies.** As shown in Table 7, individual strategies show limited effectiveness: LaserMix (45.4%), CosMix (45.1%), and PolarMix (44.8%) each capture different domain-invariant features but with performance gaps. Dual combinations demonstrate complementary benefits: CosMix+LaserMix (46.0%), PolarMix+LaserMix (45.9%), and CosMix+PolarMix (45.7%) all outperform individual approaches. The integration of all three strategies achieves optimal performance (46.2% mIoU), confirming that our BeyondMix++ successfully leverages the diverse structural priors and strengths of each mixing method to capture a more comprehensive range of domain-invariant features for robust LiDAR semantic segmentation.

# 5 Conclusion

BeyondMix addresses fundamental limitations in LiDAR segmentation domain adaptation by integrating three previously overlooked structural priors—permutation invariance, local consistency, and geometric consistency—with Mamba's sequential processing. Our approach overcomes conventional receptive field constraints while effectively modeling long-range dependencies, achieving state-of-the-art performance across benchmarks and highlighting the critical importance of structural priors in cross-domain representation learning.

**Acknowledgements**

This work was partially supported by the National Key R&D Program of China (Grant No. 2024YFB3909902), and the Youth Innovation Promotion Association of the Chinese Academy of Sciences (CAS).

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

# A Theoretical Insight

In our main text, we discuss that within the context of unsupervised domain adaptation, when a point's receptive field spans across two domains in the mixed images generated by the mix operation, the model can effectively learn domain-invariant features for this point. Conversely, if a point's receptive field is confined entirely within a single domain, the model struggles to learn domain-invariant features for that point. We now provide a detailed proof to support this claim.

We formulate our unsupervised domain adaptation framework as follows. We define the source domain $\mathcal{X}_s$ and target domain $\mathcal{X}_t$, with a shared feature space $\mathcal{Z} \subseteq \mathbb{R}^d$. Our model architecture consists of two key components: a feature extractor $f : \mathcal{X} \rightarrow \mathcal{Z}$ that maps input data to the feature space, and a classifier $h : \mathcal{Z} \rightarrow \mathcal{Y}$ that performs the classification task. The learning objective combines two loss functions: the cross-entropy loss $\mathcal{L}_{\text{CE}}$ for supervised learning on the source domain, and the domain alignment loss $\mathcal{L}_{\text{align}}$ to minimize the discrepancy between feature distributions of the source and target domains. For $x_s \in \mathcal{X}_s$ and $x_t \in \mathcal{X}_t$, generate a mixed sample:

$$x_{mix} = \lambda x_s + (1 - \lambda)x_t, \quad \lambda \sim \text{Beta}(\alpha, \beta), \tag{15}$$

where $\lambda \sim \text{Beta}(\alpha, \beta)$ indicates that the random variable $\lambda$ follows a Beta distribution with parameters $\alpha$ and $\beta$.

We prove two claims: **1. Forward Proposition:** If a receptive field spans both domains (*i.e.,* covers regions from both $x_s$ and $x_t$, domain-invariant features are learned. **2. Reverse Proposition:** If a receptive field covers only a single domain, domain-invariant features cannot be effectively learned.

**1. Proof of Forward Proposition: Cross-Domain Receptive Fields Enable Domain-Invariant** Features

For a mixed sample, the gradient of the loss *w.r.t.* feature extractor parameters $\theta_f$ is:

$$\frac{\partial \mathcal{L}_{\min}}{\partial \theta_f} = \lambda \frac{\partial \mathcal{L}_{\text{CE}}(h(f(x_s)), y_s)}{\partial \theta_f} + (1 - \lambda) \frac{\partial \mathcal{L}_{\text{align}}(f(x_t), f(x_s))}{\partial \theta_f}. \tag{16}$$

We observe that the first term ($\lambda$-weighted) enforces sensitivity to source domain labels, while the second term ($(1 - \lambda)$-weighted) enforces alignment between target and source features. The combined gradient forces $\theta_f$ to satisfy constraints from both domains, thereby driving the learning of domain-invariant patterns. This can be understood through an information-theoretic lens: maximizing cross-domain mutual information $I(z; y)$ while minimizing $I(z; d)$ (where $d$ denotes the domain label). The mix operation effectively disrupts domain-specific information, leading to:

$$I(z; d) \leq H(d) - H(d|z) \approx 0, \tag{17}$$

where $z$ implies domain-invariant representations. Therefore, cross-domain receptive fields enable the model to learn domain-invariant features.

**2.Proof of Reverse Proposition: Single-Domain Receptive Fields Fail to Learn Domain-Invariant Features**

If $\lambda \in \{0, 1\}$ (no mixing), gradients depend solely on one domain:

$$\frac{\partial \mathcal{L}}{\partial \theta_f} = \begin{cases} \frac{\partial \mathcal{L}_{\text{CE}}(h(f(x_s)), y_s)}{\partial \theta_f}, & \lambda = 1, \\ \frac{\partial \mathcal{L}_{\text{align}}(f(x_t))}{\partial \theta_f}, & \lambda = 0. \end{cases} \tag{18}$$

**Source-only training** ($\lambda = 1$)**:** Overfitting to domain-specific features (e.g., textures) degrades target generalization.

**Target-only training** ($\lambda = 0$)**:** Lack of supervision leads to trivial alignment (*e.g.,* feature collapse).

Due to domain shift $(P(y|x_s) \neq P(y|x_t))$, single-domain training causes:

$$f(x_s) \neq f(x_t) \quad \text{even if} \quad y_s = y_t, \tag{19}$$

which results in misaligned distributions. The Maximum Mean Discrepancy (MMD) between domains remains:

$$\text{MMD}(\mathcal{Z}_s, \mathcal{Z}_t) = \|\mathbb{E}_{z_s}[\phi(z_s)] - \mathbb{E}_{z_t}[\phi(z_t)]\|_{\mathcal{H}} > 0, \tag{20}$$

where $\phi$ maps to a reproducing kernel Hilbert space. Therefore, single-domain receptive fields fail to learn domain-invariant features.

# B More Related Works

**Point Cloud Semantic Segmentation**. Point cloud semantic segmentation has evolved through three primary representation paradigms: (1) Projection-based approaches like MVCNN [51] and RangeNet++ [33] utilize

CNN architectures via 2D mapping but sacrifice geometric detail; (2) Voxel-based methods improve computational efficiency through regular grid processing, as demonstrated by MinkowskiNet [5], TANet [31], and Cylinder3D [80]; and (3) Point-based networks enable end-to-end processing of raw point clouds, progressing from PointNet [42] through PointNet++ [43], DGCNN [63], Point Transformer [76], and PointNeXt [44], each enhancing local geometric modeling capabilities. This work employs MinkowskiNet as its backbone architecture due to its optimal balance between computational efficiency and segmentation accuracy.

**State Space Models**. State Space Models (SSMs) have evolved into powerful sequence modeling tools, progressing from S4 [10], which established HiPPO-based long-range dependency modeling, through computational refinements in S4D [11], DSS [9], and S5 [49], culminating in Mamba [8] with its data-dependent state dynamics and hardware-aware design. In computer vision, SSMs have demonstrated remarkable versatility: Vision Mamba [30] pioneered pure SSM architectures for image tasks, VM-UNet [47] adapted them for medical image segmentation, while S4ND [35] and Motion Mamba [75] extended their application to multi-dimensional data and action recognition. Recent advances including PointMamba [28], PCM [74], and Mamba3D [12] have successfully adapted Mamba to unordered point cloud data. Our work further extends this trajectory by demonstrating SSMs' effectiveness in unsupervised domain adaptation for LiDAR 3D point clouds, outperforming previous approaches in addressing domain shift challenges.

**Multi-modal UDA methods for Point Cloud Segmentation**. Unlike previous 2D unsupervised domain adaptation approaches [4, 26, 27, 55, 3, 24, 25] for segmentation tasks [54, 52, 53], unsupervised domain adaptation (UDA) for 3D semantic segmentation has garnered significant scholarly attention, particularly regarding the principled exploitation and fusion of multimodal information streams for cross-domain generalization. Seminal investigations [21, 60] explored the utilization of auxiliary modalities such as depth—available exclusively during source domain training—to facilitate adaptation of 3D semantic segmentation frameworks. Subsequently, xMUDA [14] established a paradigmatic approach by enforcing cross-modal consistency constraints, thereby enabling bidirectional knowledge transfer between image and point cloud representations to enhance domain generalization capabilities. Further advancements to this paradigm [15] incorporated sophisticated cross-modal fusion mechanisms and contrastive learning objectives to optimize representation alignment across both modalities and domains. Contemporary research has extended these foundational approaches through integration of advanced vision architectures and refined fusion methodologies. For instance, [40] incorporates the Segment Anything Model [18] (SAM) to augment 2D modality representations, consequently enhancing 3D segmentation performance through more effective cross-modal knowledge transfer. The authors of [67] propose a sequential fusion-then-distillation framework, which first aligns 2D and 3D feature representations within a shared latent manifold before employing positive distillation techniques to preserve complementary modality-specific information during the adaptation process. Furthermore, [50] introduces an adaptive regularization framework for modality-guided feature fusion, facilitating dynamic and contextually appropriate integration of visual and geometric information under domain distribution shifts.

# C  More Implementation Details

Our implementation framework leverages PyTorch [38] and MinkowskiEngine [5], running on a single NVIDIA RTX A6000 GPU with 48GB VRAM. The semantic segmentation backbone employs a U-Net architecture MinkUNet34. For structural priors and Long-Range dependencies, we adhered to the default configuration with the Mamba architecture parameterized as follows: input dimension of 256, hidden state dimension of 256, convolutional kernel width of 4, and expansion factor of 2. In alignment with established practices in recent literature [48, 72, 73], we utilize raw $XYZ$ coordinates as the primary input features. The critical hyperparameters $\alpha$ and $val$ are configured at 0.99 and 100, respectively. Our processing pipeline maintains a consistent voxel size of 0.05m while accommodating variable point cloud densities without explicit size constraints. The training regimen incorporates a comprehensive suite of data augmentation techniques, including random rotational transformations, Gaussian noise perturbation, coordinate jittering, and other enhancement strategies. The optimization process employs the Adam optimizer [17] with a starting learning rate of 2.5e-4, governed by a polynomial decay schedule with power coefficient 0.9. The network undergoes training for 100,000 iterations with each batch containing 2 samples. The loss function coefficients $\lambda_{mix}$ and $\lambda_{prior}$ are set to 1 and 0.01, respectively. The downsampling rate $\rho$ is configured at 50%, while the number of cylindrical partitions $L$ and rectangular partitions $V$ are set to 3 and 5, respectively. When implementing BeyondMix++ with CosMix integration, which originally generates bidirectional mixed domains (source→target and target→source), we randomly select a single directional transformation for intermediate domain generation to maintain computational efficiency while preserving adaptation benefits.

# D  Space-filling Curve.

To more intuitively observe the impact of different space-filling curves on spatial proximity, we visualize the Hilbert curve and Z-order curve as shown in Figure 4 (a), (b), (e) and (f) with dimensions of three and two, respectively. Z-order curves are renowned for their computational efficiency, while Hilbert curves are

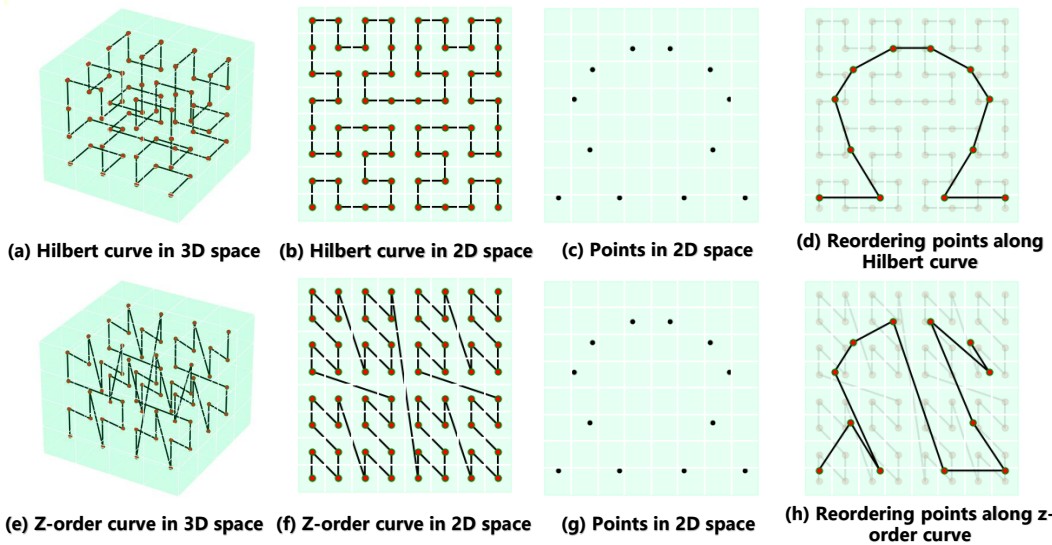

Figure 4: Space-filling curves.

distinguished by their locality-preserving property. For clarity of demonstration, we present our analysis in 2D. The spatial proximity in point clouds can be effectively preserved through point cloud serialization, whereby points that are adjacent in sequences maintain their neighborhood relationships in the point cloud representation. Figure 4 (c) and (g) illustrate randomly distributed points, which are subsequently sorted using the Hilbert curve and Z-order curve as shown in Figure 4 (d) and (h). As observed, and consistent with previous research findings [36], the Hilbert curve demonstrates superior spatial proximity preservation compared to the Z-order curve.

# E   Pseudo Algorithm

Algorithm 1 presents the comprehensive training procedure of our proposed BeyondMix++ framework for LiDAR semantic segmentation under unsupervised domain adaptation. The algorithm operates on a teacher-student architecture (self-training paradigm), processing both source domain data with ground truth labels and unlabeled target domain data. For each training batch, we first compute supervised segmentation loss on source data and adaptation loss using teacher-generated pseudo-labels on target data. The core of BeyondMix++ is constructing three additional structural priors: (1) **Permutation Invariance Prior**, whereby point cloud representations should remain consistent regardless of acquisition trajectory or scanning order [42, 43] (e.g., different angular perspectives or sampling paths), preserving invariance to permutation operations. (2) **Local Consistency Prior**, whereby point cloud features should maintain consistency across different local spatial partitions [63, 23], independent of acquisition perspective or artificially defined spatial segmentation schemes; and (3) **Geometric Consistency Prior**, whereby LiDAR point cloud geometric structures (surface curvatures, normal vectors) should maintain stability under various processing operations [29, 71], with remaining points preserving critical geometric information even under partial masking. However, these three structural priors intuitively resist straightforward implementation through mix-based paradigm like previous work. These components collectively form the total loss, which works alongside traditional mixing loss to guide the student network's training. The teacher network parameters are periodically updated through exponential moving average (EMA) of the student network weights, ensuring stable knowledge transfer. This unified approach effectively leverages Mamba to fully explore and utilize structural prior cues and expand receptive fields and enhance the ability to extract domain-invariant representations from unstructured 3D LiDAR point cloud data.

# F   More Metrics

We adopt the widely accepted Intersection over Union (IoU) metric for semantic segmentation assessment. The IoU is calculated separately for each semantic category, providing class-specific insight into segmentation quality. To obtain a holistic view of model performance, we derive the Mean Intersection over Union (mIoU) by averaging these individual class IoU values. This comprehensive measurement effectively captures both the precision and consistency of our segmentation results across the entire spectrum of semantic categories.

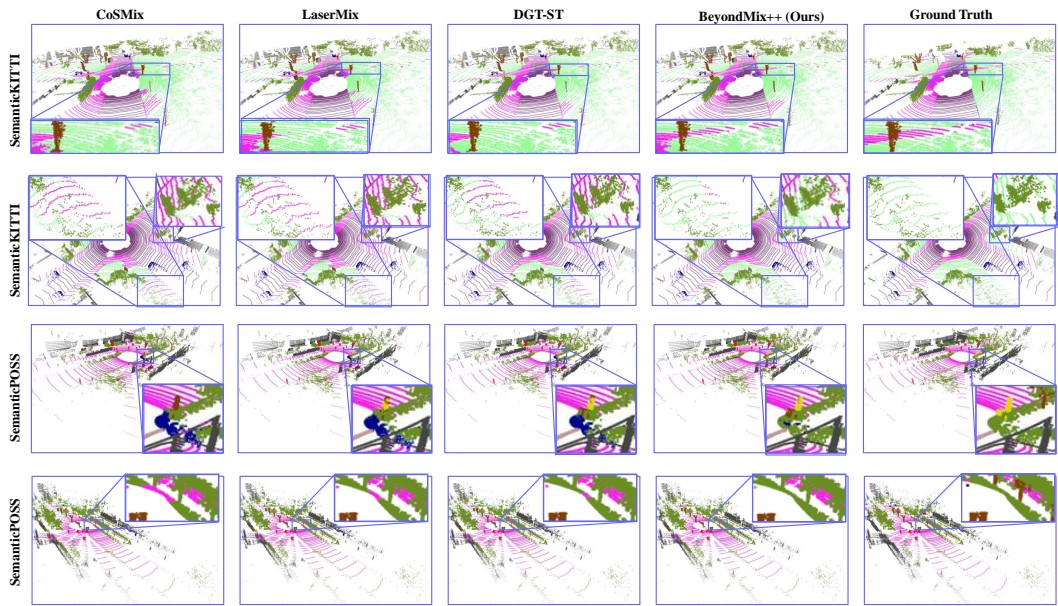

Figure 5: Qualitative results of UDA segmentation for SynLiDAR → SemanticKITTI and SynLiDAR → SemanticPOSS tasks. Boxes highlight some regions of interest.

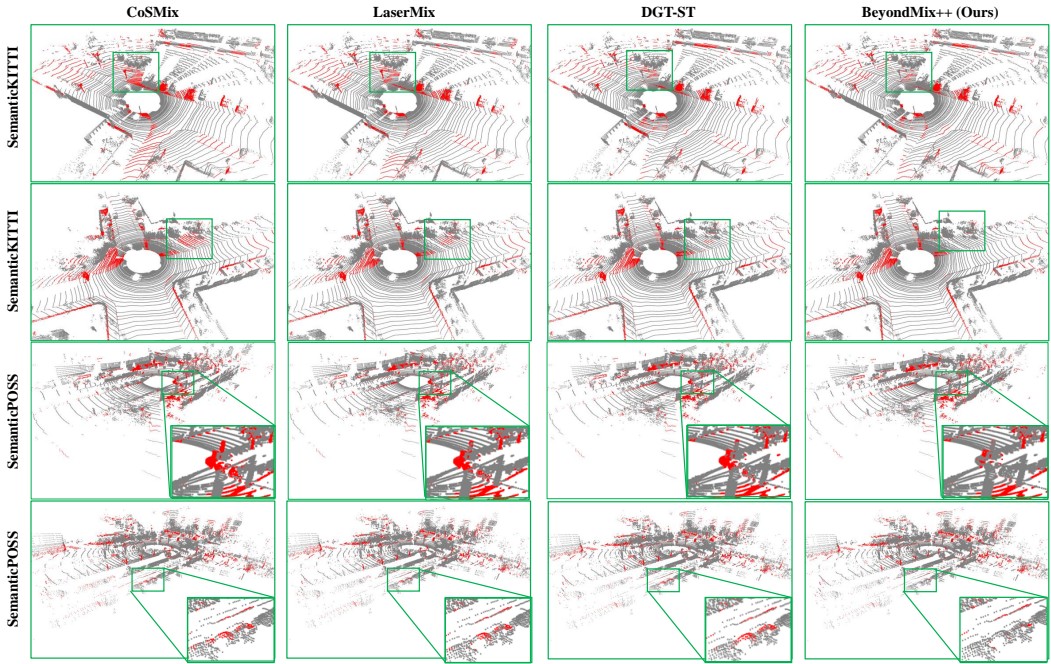

Figure 6: Error Map of UDA segmentation for SynLiDAR → SemanticKITTI and SynLiDAR → SemanticPOSS tasks. Boxes highlight some regions of interest. The binary visualization scheme employs red coloring to highlight misclassified points, whereas gray regions represent accurate predictions.

---

**Algorithm 1** Pseudo algorithms of BeyondMix++.

---

1: **Inputs:** Source Domain $\mathcal{D}^S = \{(x^s, y^s)\}$, Target Domain $\mathcal{D}^t = \{x^t\}$
2: **Define:** Student Network $f_{stu}$, Teacher Network $f_{tea}$, Momentum Coefficient $\alpha$, Update interval $val$
3: **Output:** Student Network $f_{stu}$
4: **for each** batch of $(x^s, y^s)$ in $\mathcal{D}^S$, $x^t$ in $\mathcal{D}^t$. For brevity. **do**
5:     # **Source Domain:**
6:     Calculate $\mathcal{L}^s$ for $f_{stu}$ by Eq. (1)                         ▷ Source loss
7:     # **Target Domain:**
8:     Obtain pseudo-labels from $f_t$ by Eq. (3)
9:     Calculate $\mathcal{L}^t$ for $f_{tea}$ by Eq. (2)                       ▷ Target loss
10:    # **Mixed Scan:**
11:    Calculate $\mathcal{L}_{\mathrm{mix}}$ for $f_{stu} = h \circ g$ by Eq. (7)              ▷ Mix loss
12:    # **Permutation Invariance Prior:**
13:    Calculate permutation-sensitive features $O_H$ and $O_Z$ by Eq. (8)
14:    Calculate $\mathcal{L}_{permutation}$ for $f_{stu}$ by Eq. (15)     ▷ Permutation Invariance Prior loss
15:    # **Local Consistency Prior:**
16:    Calculate permutation-sensitive features $O_L$ and $O_V$ by Eq. (8)
17:    Calculate $\mathcal{L}_{local}$ for $f_{stu}$ by Eq. (10)             ▷ Spatial Consistency loss
18:    # **Geometric Consistency Prior:**
19:    Calculate permutation-sensitive features $O$ and $\tilde{O}$ by Eq. (12)
20:    Calculate $\mathcal{L}_{geometric}$ for $f_{stu}$ by Eq. (13)          ▷ Density Consistency loss
21:    # **Training:**
22:    Calculate total loss:
23:    $\mathcal{L}_{\mathrm{total}} = \mathcal{L}^s + \mathcal{L}^t + \lambda_{mix}\mathcal{L}_{\mathrm{mix}} + \lambda_{prior}(\mathcal{L}_{\mathrm{permutation}} + \mathcal{L}_{\mathrm{local}} + \mathcal{L}_{\mathrm{geometric}})$, by Eq. (14)
24:    Gradient backward $\mathcal{L}_{\mathrm{total}}$ for $f_{stu}$             ▷ Update student model
25:    # **EMA Update:**
26:    **if** Interval == val: **then**
27:       $\theta_i^{tea} = \alpha\theta_{i-val}^{tea} + (1-\alpha)\theta_i^{stu}$, Update teacher model
28:    **end if**
29: **end for**

---

Table 8: Ablation studies of EMA coefficient $\alpha$.

| EMA $\alpha$ | mIoU(%) |
|---|---|
| 0.5 | 45.6 **(-0.6)** |
| 0.7 | 45.9 **(-0.3)** |
| 0.9 | 46.0 **(-0.2)** |
| 0.99 | **46.2** |
| 0.999 | 45.8 **(-0.4)** |

Table 9: Ablation studies of Threshold $\tau$.

| Threshold $\tau$ | mIoU(%) |
|---|---|
| 0.5 | 45.4 **(-0.8)** |
| 0.6 | 45.8 **(-0.4)** |
| 0.7 | **46.2** |
| 0.8 | 45.9 **(-0.3)** |
| 0.9 | 44.9 **(-1.3)** |

# G   More Qualitative Results

To further validate the effectiveness of our proposed BeyondMix++ framework, we present qualitative segmentation results on both source and target domains. Figure 5 illustrates representative examples that demonstrate the superior cross-domain generalization capabilities of our approach compared to baseline methods. In the first row featuring SemanticKITTI scenes, BeyondMix++ produces significantly more accurate and consistent sidewalk segmentation. The baseline method erroneously classifies portions of terrain as sidewalks, while our approach correctly delineates the boundary between these semantically similar but functionally distinct classes. The second row demonstrates BeyondMix++'s superior performance on terrain classification within SemanticKITTI. Our method correctly identifies terrain regions that are misclassified by the baseline approach, particularly in areas where terrain meets boundaries. In the SemanticPOSS visualizations (third row), BeyondMix++ achieves substantially more accurate segmentation of vegetation and other-vehicle classes. The baseline method exhibits fragmented predictions and class confusion in these categories, whereas our approach produces coherent and semantically consistent segmentations. The fourth row further validates BeyondMix++'s cross-domain generalization capabilities, where it correctly segments vegetation and sidewalk regions in SemanticPOSS that are severely misclassified by baseline approaches. The baseline exhibits notable confusion between vegetation and sidewalk—a common challenge in domain adaptation for LiDAR segmentation. These

qualitative results collectively demonstrate that BeyondMix++ effectively leverages different structural priors to learn domain-invariant features. The State Space Model architecture enhances BeyondMix++'s ability to expand receptive fields and extract robust features from unstructured 3D point cloud data. By incorporating Permutation, Local, and Geometric consistency prior mechanisms, our approach successfully captures long-range contextual information and geometric relationships that remain stable across domains, leading to more accurate and consistent segmentation across different LiDAR sensor configurations and environmental conditions.

## H    More Error Map Results

In Figure 6, we present additional visualization results with corresponding error maps for the SynLiDAR $\rightarrow$ SemanticKITTI and SynLiDAR $\rightarrow$ SemanticPOSS adaptation scenarios, comparing our BeyondMix++ approach with CoSMix [48], LaserMix [19], and DGT-ST [73] The error maps clearly demonstrate that BeyondMix++ produces significantly fewer misclassifications compared to alternative methods, underscoring the effectiveness of our proposed approach in cross-domain LiDAR semantic segmentation.

Table 10: Comparison between different coefficient $\lambda_{mix}$.

| $\lambda_{mix}$ | mIoU(%) |
|---|---|
| 0.5 | 45.5 (-0.7) |
| 0.8 | 45.8 (-0.4) |
| 1 | **46.2** |
| 1.2 | 46.1 (-0.1) |

Table 11: Comparison between different coefficient $\lambda_{uni}$.

| $\lambda_{uni}$ | mIoU(%) |
|---|---|
| 0.1 | 44.4 (-1.8) |
| 0.05 | 45.2 (-1.0) |
| 0.01 | **46.2** |
| 0.001 | 45.5 (-0.7) |

## I    More Ablation Study.

**Confidence Threshold.** Table 9 presents our ablation study on the confidence threshold $\tau$, which plays a critical role in pseudo-label generation for unsupervised domain adaptation. Our experiments demonstrate that $\tau$=0.7 achieves optimal performance (46.2% mIoU), establishing an effective balance between pseudo-label quality and quantity. Lower threshold values (0.5 and 0.6) result in decreased performance (-0.8% and -0.4% respectively), indicating that excessively lenient thresholds introduce noisy pseudo-labels that adversely affect training. Conversely, higher threshold values (0.8 and 0.9) also lead to performance degradation (-0.3% and -1.3% respectively), with a particularly significant drop at $\tau$=0.9. This suggests that overly strict thresholds discard potentially useful supervision signals from moderately confident predictions. The sharp performance decline with $\tau$=0.9 (-1.3%) is especially noteworthy, indicating that excessively high confidence thresholds severely limit the number of available pseudo-labels, significantly hampering the model's ability to adapt to the target domain. These results confirm that $\tau$=0.7 provides the optimal trade-off between pseudo-label precision and recall, enabling effective knowledge transfer in our BeyondMix++ framework for LiDAR semantic segmentation under domain shift conditions.

**EMA.** Table 8 presents our ablation study on the Exponential Moving Average (EMA) coefficient $\alpha$, which controls how quickly the teacher model incorporates updates from the student model. We observe that $\alpha$=0.99 achieves optimal performance (46.2% mIoU), establishing an effective balance between stability and adaptability in teacher-student knowledge transfer. Lower $\alpha$ values (0.5, 0.7, and 0.9) show progressively increasing performance (-0.6%, -0.3%, and -0.2% respectively), indicating that faster teacher updates compromise model stability by introducing excessive fluctuations during training. Conversely, an extremely high $\alpha$ value (0.999) also results in performance degradation (-0.4%), suggesting overly conservative teacher updates fail to effectively incorporate emerging domain knowledge. These results confirm that $\alpha$=0.99 optimally balances stability and adaptability in the teacher-student framework, enabling effective knowledge distillation for unsupervised domain adaptation in LiDAR semantic segmentation. This finding aligns with our theoretical understanding that the teacher model must maintain sufficient consistency while gradually incorporating valuable representations learned by the student model.

**Balancing Coefficients.** Table 10 shows $\lambda_{mix} = 1.0$ yields optimal performance (46.2% mIoU). Lower values (0.5, 0.8) reduce performance by 0.7% and 0.4%, suggesting insufficient mixed-domain learning, while a higher value (1.2) shows minimal impact (-0.1%). This suggests the model exhibits robustness to small increases in this coefficient, with a balanced value of 1.0 providing optimal regularization. For $\lambda_{prior}$ (Table 11), the optimal value is 0.01. Larger values (0.1, 0.05) significantly degrade performance (-1.8%, -1.0%) by overemphasizing consistency constraints, which can overshadow the primary segmentation objective. Conversely, a smaller value (0.001) also reduces performance (-0.7%) as the regularization becomes too weak to effectively guide domain-invariant feature learning. These findings confirm that proper balance between loss components ($\lambda_{mix} = 1.0$,

$\lambda_{prior} = 0.01$) is crucial for effective domain adaptation and diverse structural prior cues, as well as maximizing the extraction of domain-invariant features across different receptive fields.

**Different Long-range Modeling Methods.** To comprehensively validate our approach, we evaluated alternative long-range dependency modeling techniques, including the widely adopted Point Transformer v3 (PTv3) [66], which implements attention mechanisms within locally grouped point cloud sequences. Our analysis revealed that the standard PTv3 implementation constrains the receptive field to 1024 points, which proves insufficient for our application domain. As demonstrated in our main text, the minimum requisite receptive field exceeds 1300 points (potentially substantially more, contingent upon mixing strategies and scanning methodologies). Mamba, conversely, exhibits an unbounded receptive field, rendering it particularly suitable for modeling the long-range dependencies essential to our task domain. The quantitative experimental results presented in Table **??** substantiate our theoretical assertions. Furthermore, Pamba establishes that "SSM can process the entire point cloud of 100,000+ points expeditiously without subdivision and model uncompressed long-range dependencies," affirming that "SSM demonstrates superior suitability compared to transformer architectures for extracting long-term dependencies." This aligns precisely with our requirements, as each scan in our outdoor LiDAR dataset comprises over 100,000 points. While we implemented PTv3 with a parameter setting of $P = 2048$, the quadratic computational complexity inherent to transformer architectures rendered the processing speed prohibitively inefficient for practical deployment.

Table 12: Comparison between Different Long-range Modeling Methods.

| Method | mIoU (%) |
|---|---|
| PTv3 (P=1024) | 45.6 |
| Mamba | 46.2 |

## J   Limitations

Despite the demonstrated effectiveness of BeyondMix, several limitations warrant acknowledgment. First, the sequential transformation of 3D point clouds introduces computational overhead that may impact real-time performance in resource-constrained autonomous systems. Second, while our approach leverages three key structural priors, additional domain-specific priors may exist that could further enhance performance. Third, the effectiveness of our method may vary across different sensor configurations and point cloud densities, potentially requiring domain-specific adjustments. Finally, our current implementation primarily focuses on semantic segmentation tasks, and its generalizability to other point cloud understanding tasks, such as object detection or instance segmentation, requires further investigation.

## K   Broader Impacts

BeyondMix contributes to advancing autonomous driving technologies by enhancing LiDAR semantic segmentation capabilities across diverse environmental conditions. This advancement has potential societal benefits, including improved transportation safety, reduced accidents, and enhanced mobility for individuals with disabilities. However, the deployment of such technologies raises important ethical considerations regarding privacy, security, and the potential for algorithmic bias. Additionally, as autonomous systems become more prevalent, careful consideration must be given to their environmental impact, including energy consumption and electronic waste. We encourage the research community to address these broader implications while advancing the technical capabilities of domain adaptation methods for 3D perception systems.

