# OpenReview forum: "BeyondMix: Leveraging Structural Priors and Long-Range Dependencies for Domain-Invariant LiDAR Segmentation"
_NeurIPS.cc/2025/Conference — NeurIPS 2025 poster_

### Official Review · Reviewer_6p4o · 2025-06-30

**Clarity:** 3
**Significance:** 3
**Originality:** 3
**Rating:** 4
**Confidence:** 4

**Summary:**

This paper addresses unsupervised domain adaptation (UDA) for LiDAR semantic segmentation by improving upon existing mix-based strategies. The proposed method, BeyondMix, introduces a framework that explicitly models three structural priors: permutation invariance, local consistency, and geometric consistency. These priors are used to preserve intrinsic structure in point cloud data across domains. To enhance the model’s ability to capture long-range dependencies, BeyondMix leverages State Space Models, specifically Mamba, applied to sequentially ordered point cloud representations via space-filling curves. Experiments on SynLiDAR to SemanticKITTI and SemanticPOSS benchmarks show that BeyondMix achieves state-of-the-art cross-domain segmentation performance compared to prior methods.

**Questions:**

1. What are the runtime and memory overheads introduced by BeyondMix, particularly due to Mamba and the multiple auxiliary losses?
2. Have you tested or considered BeyondMix under real-to-real domain shifts (e.g., SemanticKITTI → SemanticPOSS)? Would the same priors and Mamba integration apply effectively?
3. Could you report per-class IoU in the ablation studies? This would help determine whether specific object categories benefit more from certain structural priors.
4. Could you clarify how Mamba is integrated into your architecture (e.g., at voxel level or raw sequence)? Additionally, how does your use of Mamba compare in motivation and performance to recent long-range modeling methods such as PointTransformer v3? A conceptual and empirical contrast would help clarify the choice.

**Ethical Concerns:**

["NO or VERY MINOR ethics concerns only"]

**Final Justification:**

The rebuttal has satisfactorily addressed my main concerns regarding computational overhead, real-world domain shift performance, and the rationale for using Mamba. The clarifications, together with the additional per-class ablation results, have resolved my doubts about the effectiveness and generalization ability of the proposed method. I therefore maintain my positive recommendation.

**Limitations:**

yes

**Quality:**

3

**Strengths And Weaknesses:**

**Strengths:**
1. The paper precisely identifies the limitations of existing mix-based UDA methods for LiDAR segmentation, particularly their underutilization of structural priors and limited receptive fields. It then formulates a method that directly addresses these issues through principled design.
2. BeyondMix introduces three underexplored structural priors of permutation invariance, local consistency, and geometric consistency formalized as loss functions. These are combined with a state-space sequence model (Mamba) using space-filling curves, enabling long-range context modeling across unordered point clouds.
3. The method achieves state-of-the-art results on two challenging UDA benchmarks (SynLiDAR → SemanticKITTI and SemanticPOSS), with significant mIoU gains across multiple semantic classes and domains.
4. The paper includes extensive ablations showing the individual and combined benefits of each structural prior and Mamba integration, supporting the effectiveness and complementarity of its components.

**Weaknesses:**
1.  The method involves non-trivial additions (Mamba, multiple loss terms), but the paper does not report training/inference time or memory usage, leaving concerns about practical efficiency unaddressed.
2.   All experiments are from synthetic to real. The paper does not examine real-to-real domain adaptation or more diverse shifts (e.g., different sensor types), which narrows its generalization claims.
3.  The ablation study reports only overall mIoU, without showing per-class performance. This prevents analysis of which semantic classes benefit from each structural prior.
4. The main text lacks sufficient discussion of alternative long-range modeling methods, particularly recent transformer-based architectures like PointTransformer v3. These are only briefly mentioned in the appendix. The absence of conceptual or empirical comparison weakens the justification for using Mamba and diminishes clarity on architectural novelty.

---

> ### Author Rebuttal · Authors · 2025-07-30
>
> Thank you for your thorough review and thoughtful assessment of our manuscript. We appreciate your recognition that our work, BeyondMix, **successfully identifies the fundamental limitations of existing mix-based UDA methods**, provides **comprehensive ablation studies**, and achieves **state-of-the-art performance metrics.**
>
> We acknowledge your primary concerns regarding the need for further clarification of certain aspects of our methodology. In the following response, we address each of your comments systematically to provide the requested explanations and clarifications. **Please feel free to use the discussion period if you have any additional questions.**
>
> ---
>
> ## Q1:
> What are the runtime and memory overheads introduced by BeyondMix, particularly due to Mamba and the multiple auxiliary losses?
>
> ## A1:
> Below, we provide a comprehensive analysis of computational requirements including GPU memory utilization, per-iteration runtime, and FLOPs for both training and inference phases.
>
> During the **training stage**, the incorporation of long-range modeling mechanisms inevitably introduces additional computational overhead. However, this increase remains **relatively modest** due to the **near-linear complexity (approximately O(n))** of our Mamba-based implementation.
>
> **For inference** , our method employs **a standard network architecture without auxiliary modules**, resulting in computational requirements comparable to baseline approaches.
>
> As demonstrated in Table R1, BeyondMix++ exhibits **identical GPU memory consumption, per-iteration runtime, and FLOPs during inference** compared to baseline methods such as LaserMix and DGT-ST. Notably, our approach achieves substantial performance improvements (+10.2% over LaserMix and +3.1% over DGT-ST) while incurring only **acceptable additional training overhead**, highlighting the favorable efficiency-performance trade-off of our proposed framework.
>
> **Table R1: Quantitative Analysis of Computational Overhead**
>
> | Method | GPU Memory (Train/Infer, GB/GB) | Time per Iteration (Train/Infer, iteration/s, scan/s) | FLOPs (Train/Infer, G/G) |
> |-|-|-|-|
> | **LaserMix** | 17.81/2.5 | 0.92/6.96 | 65.10/32.95 |
> | **DGT-ST** | 14.72/2.5 | 0.95/6.95 | 63.68/32.95 |
> | **BeyondMix++（Ours）** | 22.39/**2.5** | 0.67/**6.95** | 76.84/**32.95** |
>
> ---
>
> ## Q2:
> Are you tested or considered BeyondMix under real-to-real domain shifts (e.g., SemanticKITTI → SemanticPOSS)? Would the same priors and Mamba integration apply effectively?
>
> ## A2:
> In accordance with established research paradigms [CosMix, PolarMix, LaserMix, DGT-ST], we construct UDA frameworks spanning synthetic-to-real scenarios (Syn→SK, Syn→SP), capitalizing on the availability of synthetic annotations. This virtual-to-real generalization framework aligns more closely with practical deployment scenarios. Furthermore, to substantiate the generalization of our methodology, we present **real-to-real adaptation results (SP→SK)** in Table R2. For **cross-modal sensing evaluation**, we establish an SK (rotating LiDAR with 64 beams)→PGT configuration, leveraging the PandarGT (PGT, a solid-state front-facing LiDAR with 150 beams) from Pandaset. As empirically demonstrated in Table R2, our approach consistently attains state-of-the-art performance metrics. This uniform superiority across synthetic-to-real, real-to-real, and cross-sensor domain adaptation paradigms corroborates the robust generalization capacity of our proposed methodology.
>
> **Table R2: Comparison Results of Other Settings**
>
> |Setting/Methods| **Source only** | **Target only** | **LaserMix** | **DGT-ST** | **BeyondMix++ (Ours)** |
> |-| -|- |-|-|-|
> | SemanticPOSS → SemanticKITTI (SP→SK) |22.5| 60.4| 29.5 | 38.1| **42.0**|
> | SemanticKITTI → PandarGT (SK→PGT) |27.5|78.2| 49.2| 52.5| **60.7**|
>
> ---
>
> ## Q3:
> Could you report per-class IoU in the ablation studies? This would help determine whether specific object categories benefit more from certain structural priors.
>
> ## A3:
> As requested, we provide a comprehensive per-class performance analysis from our ablation study in Table R3.
>
> Upon incorporating $L_{permutation}$, we observe significant improvements in the "sidewalk", "road", "person", and "pole" categories. This enhancement can be attributed to the distinct characteristics of our space-filling curves. The Hilbert curve constructs a continuous path traversing 3D voxels with strong locality preservation, resulting in more comprehensive modeling of relatively small local objects such as "person" and "pole". Conversely, the Z-curve's horizontal continuity properties offer advantages in modeling planar categories like "sidewalk" and "road".
>
> When $L_{local}$ is introduced, substantial performance gains emerge in the "car", "motorcycle", "vegetation", "road", and "building" categories. This aligns with our local consistency structural priors, where cylindrical partitioning recognizes that objects proximal to LiDAR sensors typically include "car" and "motorcycle" instances, while distant points frequently represent "vegetation". Additionally, rectangular partitioning effectively preserves local features by leveraging the observation that "building" structures typically appear at higher elevations while "road" surfaces are situated at lower positions. These experimental results corroborate our theoretical framework.
>
> The addition of $L_{geometric}$ yields balanced improvements across all categories, indicating that geometric consistency across varying point densities serves as a universal constraint applicable to diverse semantic classes.
>
> **Table R3: Ablation Study of Per-class Performance.**
>
> |Methods|mIoU|car|bi.cle| mt.cle | truck | oth-v. | pers. | bi.clst | mt.clst | road | parki. | sidew. | other-g. | build. | fence | veget. | trunk | terr. | pole | traf. |
> |--|--|-|-|-|-|-|-|-|-|-|-|--|--|-|-|-|--|-|--|-|
> | $BaseLine$ |43.1 |92.9 |17.3|43.4|15.0 |6.1|49.2 |54.2 |4.2|86.4 |19.1 | 62.3 | 0.0 | 78.2 | 9.2 | 83.3 | 56.0 | 59.1 | 51.2 | 32.3 |
> | $+L_{mix}$ |44.1 |92.4 |17.9|42.9| 16.6 |13.5|52.1|53.9|6.6|86.3 |21.5 | 59.9 | 0.0 | 77.9 | 14.4 | 79.8 | 57.5 | 59.7 | 52.4 | 32.6 |
> | $+L_{mix}+L_{permutation}$ |44.5|92.8 |17.5| 44.8 |16.1|10.2 |54.5 | 55.7 | 5.5 | 87.1 | 21.1 | 66.4 | 0.0 | 78.0 | 13.2 | 80.4 | 57.7 | 60.5 | 52.9 | 31.2 |
> | $+L_{mix}+L_{local}$ | 44.7 | 93.9|17.4 |47.2|17.3|10.7|53.6| 55.9|6.7| 87.2 | 20.7 | 65.3 | 0.0 | 79.1| 10.3|80.9| 58.3 | 61.1 | 52.5 | 30.6 |
> | $+L_{mix}+L_{geometric}$ |43.4|92.6|16.9 | 43.1|15.9 |8.8| 49.5| 54.6|5.6| 86.1 | 20.0 | 63.2 |0.0|77.5|9.3|79.9| 57.4| 60.4 | 52.7 | 30.9 |
> | $+L_{mix}+L_{permutation}+L_{local}$ |45.4 | 93.9 |15.1| 46.2 |16.7|13.3| 55.5 | 57.2 | 6.6 |86.5| 24.0| 68.0 |0.0| 78.8| 14.1 | 81.7 | 59.3 | 61.2 | 53.2 | 32.1 |
> | $+L_{mix}+L_{local}+L_{geometric}$ |45.8| 94.4| 15.2 |46.9| 17.1 |15.7| 54.9 | 59.8 | 6.7 | 86.3| 22.4| 67.4 |0.0| 79.2| 13.7 | 82.1 | 60.6 | 62.3 | 53.0 |32.0|
> | $+L_{mix}+L_{permutation}+L_{geometric}$ |45.0|93.7|14.8| 45.5 | 16.9| 15.2 | 54.7 | 57.0 |6.9| 85.9| 22.3 |67.9| 0.0| 78.6 | 11.9 | 80.4 | 60.5 | 59.2 | 52.0 | 31.9 |
> | $+L_{mix}+L_{permutation}+L_{local}+L_{geometric}$ | 46.2 | 94.0 |15.1 | 47.2 | 17.6 | 16.5 |55.2|59.9|6.8 |87.0| 24.1 | 69.3 | 0.0 | 79.3 | 14.7 | 81.8 | 61.0 | 62.8 | 53.3 | 32.1 |
>
> ---
>
> ## Q4:
> Could you clarify how Mamba is integrated into your architecture (e.g., at voxel level or raw sequence)? Additionally, how does your use of Mamba compare in motivation and performance to recent long-range modeling methods such as PointTransformer v3? A conceptual and empirical contrast would help clarify the choice.
>
> ## A4:
> We appreciate your inquiry and would like to provide a detailed explanation:
>
> To effectively model long-range dependencies, we deploy Mamba within the feature space generated by the segmentation model's encoder. This feature space offers significant advantages: reduced spatial sizes, higher dimensionality, and semantically enriched representations. By serializing these features and processing them through the Mamba architecture, we achieve alignment between corresponding features from different sequential orderings. This approach **facilitates the encoder's acquisition of structural priors, thereby enhancing segmentation performance in the target domain.**
>
> Regarding your suggestion about **Point Transformer v3 (PTv3), which implements attention mechanisms within locally grouped point cloud sequences**, our analysis reveals that the standard PTv3 implementation possesses a receptive field limited to 1024, which proves insufficient for our application. As demonstrated in line 71 of our manuscript, the minimum requisite receptive field exceeds 1300 (potentially considerably more, contingent upon mixing strategies or scanning methodologies). However, Mamba exhibits an unbounded receptive field, rendering it particularly suitable for modeling long-range dependencies essential to our task domain.  As presented in Table R4, our theoretical assertions are substantiated by quantitative experimental results.
> Concurrently, the Pamba study [1] illustrates in Figure 1 that Mamba's effective receptive field substantially surpasses that of PTv3.
>
> Furthermore, Pamba [1] establishes that "SSM can process the whole 100,000+ points expeditiously without subdivision and model uncompressed long-range dependencies," asserting that "SSM is more suitable than transformer for extracting long-term dependencies." This aligns precisely with our requirements, as each scan in our outdoor LiDAR dataset comprises over 100,000 points. While we experimented with PTv3 (P=2048), the quadratic computational complexity inherent to transformer architectures rendered the processing speed prohibitively inefficient for practical implementation.
>
> [1] Pamba: Enhancing Global Interaction in Point Clouds via State Space Model. AAAI2025
>
> **Table R4: Comparison between Different Long-range Modeling Methods.**
> |Method |mIoU (%)|
> |-| -|
> | PTv3（P=1024）| 45.6|
> | Mamba| **46.2** |

---

> > ### Comment · Reviewer_6p4o · 2025-08-04
> >
> > Thank you for addressing my comments. I have no further questions. I believe the table presented in the rebuttal would strengthen the paper and recommend including it in the final version.

---

> > > ### Author Response · Authors · 2025-08-05
> > >
> > > Dear Reviewer  6p4o,
> > >
> > > We sincerely thank you for your constructive feedback and for acknowledging our rebuttal. Your thoughtful comments helped us better communicate the key contributions of our work and further improve the overall clarity of the paper.
> > >
> > > We truly appreciate your suggestion regarding the final version, and we will incorporate the relevant materials accordingly. Your support and engagement have been very encouraging, and we are grateful for the time and care you devoted to reviewing our submission.
> > >
> > > Thank you again for your valuable feedback!
> > >
> > > Best regards,
> > >
> > > Authors of Submission 16945

---

### Official Review · Reviewer_H4Er · 2025-07-01

**Clarity:** 3
**Significance:** 3
**Originality:** 3
**Rating:** 5
**Confidence:** 3

**Summary:**

This paper introduce BeyondMix, which harnesses the capabilities of State Space Models (specifically Mamba) to construct and exploit these structural priors while modeling long-range dependencies. It tries to transcend the limited receptive fields of conventional voxel-based approaches. The suggested method has been validated to be effective.

**Questions:**

1. The Mamba model in the paper is only used as an auxiliary supervision module during the training phase. Has consideration been given to directly decoding Mamba features for classification during inference? Will this design have better results.
2. The receptive field at point B in Figure 2 contains two types of domains, but line 76 in the text states that they are not included;

**Ethical Concerns:**

["NO or VERY MINOR ethics concerns only"]

**Final Justification:**

Thanks for the authors' rebuttal. I haven't found any evidence of rating reduction and I keep the initial positive rating.

**Limitations:**

The authors discussed their limitations of their work.

**Quality:**

3

**Strengths And Weaknesses:**

**Strengths**
1. This paper introduce BeyondMix, which leverages Mamba to construct structural priors beyond the capabilities of existing mix-based methods while simultaneously exploiting long-range dependency modeling to transcend limited voxel receptive fields for enhanced domain-invariant representation learning.
2. This paper introduces the Mamba model and solves the problem of different feature expressions caused by uneven LiDAR density to a certain extent by constructing geometric consistency priors.
**Weaknesses**
1. The permutation invariant prior module only selects two specific sorting sequences and learns to make their representations in the Mamba space approach each other, rather than making their representations similar in any sorting. Therefore, whether it can be called "permutation invariant" remains to be considered.
2. The meaning of figures 1 (b) and (c) is unclear, especially the table in (c) is difficult to understand.

---

> ### Author Rebuttal · Authors · 2025-07-30
>
> We greatly appreciate your time and thoughtful assessment of our manuscript. We are particularly grateful for your recognition that our BeyondMix approach effectively **addresses the challenges of cross-domain feature representation in LiDAR-based applications**.
>
> We have carefully considered your primary concerns regarding your specific areas of interest and the inadvertent errors in our manuscript. In the following sections, we address each of your comments systematically to improve the clarity and accuracy of our paper. **Please feel free to use the discussion period if you have any additional questions.**
>
> ---
>
> ## Q1:
> The permutation invariant prior module only selects two specific sorting sequences and learns to make their representations in the Mamba space approach each other, rather than making their representations similar in any sorting. Therefore, whether it can be called "permutation invariant" remains to be considered.
>
> ## A1:
>  We apologize for any confusion regarding our methodology. While numerous space-filling curve algorithms exist, we deliberately selected two of the most widely adopted implementations—H-curve and Z-curve—to generate diverse sequential representations. Our approach specifically aims to ensure that feature representations across these different scanning orders exhibit proximity in the Mamba embedding space.
> For clarity, we will revise our terminology to **"H-Z permutation invariance"** to explicitly indicate the two specific space-filling curves employed in our framework. This modification will be incorporated in subsequent versions of the manuscript to enhance precision and prevent misinterpretation.
>
> ---
>
> ## Q2:
> The meaning of figures 1 (b) and (c) is unclear, especially the table in (c) is difficult to understand.
>
> ##  A2:
> We apologize for any confusion and will address this issue in the final manuscript.
>
> In our notation, "-" indicates structural priors not previously explored in existing methods, specifically permutation invariance, local consistency, and geometric consistency. The "x" symbol denotes structural priors that fundamentally cannot be implemented through mix-based approaches.
>
> Current mix-based methodologies each leverage a single structural prior:
>
> - **Cosmix** employs **Category Semantic Integrity** by utilizing semantic category instances as the minimal mixing unit (illustrated in Fig. 1(b), first diagram)
> - **Polarmix** implements **Azimuthal Semantic Consistency** by extracting and rotationally mixing points of identical semantic classes (depicted in Fig. 1(b), second diagram)
> - **Lasermix** enforces **Spatial Region Integrity** by partitioning scenes according to inclination patterns (shown in Fig. 1(b), third diagram)
>
> These methodologies are limited by their singular focus on one structural prior and insufficient receptive field capacity.
> Our approach incorporates both mix-compatible priors and those incompatible with mixing techniques. The additional 3 structural priors we adopt are:
>
> 1. **Permutation Invariance** (Fig. 1(b), fourth diagram): Ensures consistency across different scanning orders representing varied observational perspectives (e.g., top-down versus bottom-up)
> 2. **Local Consistency**(Fig. 1(b), fifth diagram): Leverages statistical partitioning. For instance:
>    - Cylindrical partitioning recognizes proximity-based semantic patterns (objects near sensors are typically" car"or "motorcycle"  while distant points often represent "vegetation")
>    - Rectangular partitioning utilizes Z-axis coordinates (lower values correspond to" road" while higher values indicate "buildings")
> 3. **Geometric Consistency** (Fig. 1(b), sixth diagram): Maintains consistent geometric representations across varying LiDAR point densities.
>
> Additionally, our method models long-range dependencies, substantially expanding the receptive field and addressing the domain-invariant feature learning limitations in unsupervised domain adaptation caused by restricted receptive fields.
>
> ---
>
> ## Q3:
> The Mamba model in the paper is only used as an auxiliary supervision module during the training phase. Has consideration been given to directly decoding Mamba features for classification during inference? Will this design have better results.
>
> ## A3:
> We acknowledge the suggestion regarding direct decoding of Mamba output features. As empirically demonstrated in Table R1, this approach induces substantial performance degradation. Our comprehensive analysis elucidates several contributing factors:
>
> During the training phase, our architecture leverages Mamba to process intermediate-layer features, establishing structural priors and effective receptive fields that constrain our encoder toward learning more discriminative and domain-invariant representations. However, Mamba transforms these intermediate features into an alternative high-dimensional alignment space that is fundamentally incongruent with the original feature manifold.
>
> **This transformed representation space lacks decoder-specific optimization, as the decoder was not trained to process features from this alternative embedding space.** Consequently, the semantic decoding process fails to effectively interpret these representations, resulting in substantial performance deterioration.
>
> These empirical findings substantiate that our architectural design—utilizing Mamba as a constraint mechanism during training rather than as a feature extractor during inference—constitutes the optimal configuration for cross-domain feature learning in LiDAR semantic segmentation.
>
> **Talbe R1: Comparative Analysis of Decoding Features.**
> | Decoding Features|mIoU (%)|
> | -- | -------- |
> | MAMBA-Processed Features       | 34.8     |
> | Original Intermediate-layer Features | **46.2** |
>
>
> ---
>
> ## Q4:
>  The receptive field at point B in Figure 2 contains two types of domains, but line 76 in the text states that they are not included
>
> ## A4:
> We sincerely apologize for the confusion caused by **our inadvertent error**. Thank you for bringing this issue to our attention. We incorrectly reversed the labels of point A and point B in Lines 64 and 76 of our manuscript.
>
> For clarification, the caption of Figure 2 contains the correct description. We will rectify this error in the subsequent version of our manuscript to ensure consistency throughout the document.

---

### Official Review · Reviewer_ymd8 · 2025-07-01

**Clarity:** 3
**Significance:** 2
**Originality:** 2
**Rating:** 3
**Confidence:** 5

**Summary:**

This paper introduces BeyondMix for UDA in LiDAR semantic segmentation. The main contribution is the utilization of a State Space Model (Mamba) to process point cloud features. By employing space-filling curves (e.g., Hilbert and Z-order curves) to serialize 3D point clouds, the method satisfies Mamba's requirement for sequential input. They also construct 3 structural priors: Permutation Invariance, Local Consistency, and Geometric Consistency. In addition, the authors propose leveraging Mamba's powerful long-range dependency modeling capabilities to capture context from across the entire point cloud scene.

**Questions:**

I would ask the authors to please provide a response addressing the weaknesses I have outlined to help resolve my concerns. If the authors can clarify these issues in their rebuttal, I will consider raising my score.

**Ethical Concerns:**

["NO or VERY MINOR ethics concerns only"]

**Final Justification:**

According to the author's rebuttal, I tend to maintain the score.

**Limitations:**

yes

**Quality:**

3

**Strengths And Weaknesses:**

Strengths
﻿
1. The paper identifies two main weaknesses in existing methods: "insufficient use of structural priors" and "limited receptive fields." To address these limitations, the paper proposes using the Mamba model to unify the solutions to both problems—modeling long-range dependencies and constructing new structural priors.
﻿
2. The paper resolves the challenge of processing unordered point clouds with Mamba by using space-filling curves and further leverages different serialization and partitioning schemes to construct priors. This design is highly innovative.

3. The paper is well-written and easy to understand. The results also demonstrate the effectiveness of the proposed method.
﻿
﻿
Weaknesses
﻿
1. To construct the various structural priors, the method requires multiple serializations of the features and repeated forward passes through the Mamba model. This may introduce considerably higher computational complexity than baseline methods. The paper lacks a quantitative analysis of runtime evaluation (e.g., inference time and FLOPs).

2. It's common for UDA methods like DACS, DAFormer, and DGT-ST to use a combination of source loss (L s) and mixed-sample loss (L mix). If L mix is already being used to adapt to the target domain, why is the separate target loss, L t, also necessary?.

3. The Mamba model captures long-range dependencies along the 1D path ("flatten" a 3D structure into a 1D sequence), but its ability to capture complex 3D spatial relationships that do not lie on this path may be weaker. This could lead to the loss of critical spatial information.

4. The method enforces "equivariance" on intermediate features by inverse-mapping the Mamba outputs back to their original order. However, the ultimate goal for segmentation is to achieve "invariance" in the final prediction.

5. The proposed Permutation Invariance and Local Consistency priors all work by altering the processing order of points and then enforcing consistency on the feature representations. It seems to be be conceptually overlapping.

---

> ### Author Rebuttal · Authors · 2025-07-30
>
> We sincerely appreciate your time and effort in reviewing our manuscript. We are particularly grateful for your recognition that our work, BeyondMix, is **"highly innovative,"** **"well-written and easy to understand,"** with **"effective results."**
>
> We acknowledge your primary concerns regarding the missing comparisons and additional issues you highlighted.  All your comments are addressed point by point in the following. **Please feel free to use the discussion period if you have any additional questions.**
>
> ---
>
> ## Q1:
> To construct the various structural priors, the method requires multiple serializations of the features and repeated forward passes through the Mamba model. This may introduce considerably higher computational complexity than baseline methods. The paper lacks a quantitative analysis of runtime evaluation (e.g., inference time and FLOPs).
>
> ## A1:
> We appreciate your suggestion regarding runtime evaluation. Below, we provide a comprehensive analysis of computational requirements including GPU memory utilization, per-iteration runtime, and FLOPs for both training and inference phases.
>
>
>
> During the **training stage**, the incorporation of long-range modeling mechanisms inevitably introduces additional computational overhead. However, this increase remains **relatively modest** due to the **near-linear complexity (approximately O(n))** of our Mamba-based implementation.
>
> **For inference**, our method employs a **standard network architecture without auxiliary modules** , resulting in computational requirements comparable to baseline approaches.
>
> As demonstrated in Table R1, BeyondMix++ exhibits **identical GPU memory consumption, per-iteration runtime, and FLOPs during inference** compared to baseline methods such as LaserMix and DGT-ST. Notably, our approach achieves substantial performance improvements (+10.2% over LaserMix and +3.1% over DGT-ST) while incurring only **acceptable additional training overhead**, highlighting the favorable efficiency-performance trade-off of our proposed framework.
>
> **Table R1: Quantitative Analysis of Computational Overhead**
>
> | Method | GPU Memory (Train / Infer, GB/GB) | Time per Iteration (Train / Infer, iteration/s, scan/s) | FLOPs (Train / Infer, G/G) |
> |----|---|---|--|
> | **LaserMix** | 17.81 / 2.5 | 0.92 / 6.96 | 65.10 / 32.95 |
> | **DGT-ST** | 14.72 / 2.5 | 0.95 / 6.95 | 63.68 / 32.95 |
> | **BeyondMix++（Ours）** | 22.39 / **2.5** | 0.67 / **6.95** | 76.84 / **32.95** |
>
> ---
>
> ## Q2:
> It's common for UDA methods like DACS, DAFormer, and DGT-ST to use a combination of source loss (L s) and mixed-sample loss (L mix). If L mix is already being used to adapt to the target domain, why is the separate target loss, L t, also necessary?.
>
> ## A2:
> Thank you for raising this important question regarding the necessity of target loss ($L_t$) in mix-based methods. We have conducted thorough investigations into this matter.
>
> While 2D UDA methods such as DACS and DAFormer employ loss functions formulated as $L_s+L_{mix}$, the discussion and empirical evaluation of mixing strategies in 2D domain are relatively limited. Given that our proposed approach falls within the 3D methodological framework, we primarily focus our investigation on 3D LiDAR-based techniques when examining domain adaptation strategies:
>
> - CosMix performs mutual mixing of source and target domains without intermediate forms, utilizing a total loss function of $L_{s->t}+L_{t->s}$ as defined in Equation 5 of the original paper.
> - PolarMix is primarily a data augmentation method without explicit specification regarding target loss utilization.
> - LaserMix, the current state-of-the-art mixing approach, incorporates a comprehensive loss function (Equation 9 of the original paper): $L_s+L_{mix}+L_{mt}$ , where $L_{mt}$ represents the target loss. **Their ablation studies in Table 3 (experiments (2) and (3) of the original paper demonstrate that removing target loss results in approximately 3% mIOU degradation.**
> - DGT-ST follows LaserMix's framework with a total loss function (Equation 15 of the original paper): $L_s+L_{mix}+L_{sac}$ , where $ L_{sac}$ is defined in Equation 12 of the original paper as essentially a target loss, albeit with unique teacher-student network input designs. **Their ablation studies in Table 5 of the original paper confirm that eliminating $ L_{sac}$ causes a 1.1% mIOU performance drop.**
>
> Our experimental results in Table R2 similarly indicate performance degradation when target loss $ L_t$ is removed from BeyondMix++. We posit that mix methods effectively learn cross-domain commonalities, while target loss $L_t$ enables focused learning of target domain-specific  3D knowledge, thereby enhancing model performance in the target domain.
>
>
> **Table R2：Comparative Analysis of Target Loss $L_t$**
>
> | Method          | mIoU (%)|
> | - | ---- |
> | W/O  $L_t$  | 45.3 |
> | W / $L_t$   | **46.2** |
>
> ---
>
> ##  Q3:
> The Mamba model captures long-range dependencies along the 1D path ("flatten" a 3D structure into a 1D sequence), but its ability to capture complex 3D spatial relationships that do not lie on this path may be weaker. This could lead to the loss of critical spatial information.
>
> ## A3:
> Your observation regarding the potential loss of 3D spatial relationships when flattening 3D structures into 1D sequences is valid. **However, our approach serves as a complementary component to existing methods rather than a replacement.** Our framework preserves the original spatial relationship learning while introducing additional structural priors that were previously underexplored.
>
> Specifically, we model long-range dependencies that prior approaches have neglected. By employing different sequential orderings, each capturing distinct spatial adjacency patterns and structural priors, our method effectively extracts domain-invariant features at individual spatial points—a critical aspect in unsupervised domain adaptation tasks.
>
> The diverse sequential representations enable the capture of complementary structural information while maintaining the integrity of the original spatial context, thereby enhancing the model's capacity to learn transferable features across domains without compromising spatial understanding.
>
> ---
>
> ## Q4:
> The method enforces "equivariance" on intermediate features by inverse-mapping the Mamba outputs back to their original order. However, the ultimate goal for segmentation is to achieve "invariance" in the final prediction.
>
> ## A4:
> Our approach can be elucidated through two principal perspectives:
>
> (1) **Intermediate feature equivariance facilitates terminal invariance.** Numerous classical paradigms, particularly consistency regularization frameworks, leverage data augmentation operations such as mixup and cropping to induce equivariance in the feature space while imposing invariance constraints in the prediction manifold. This methodology enhances model robustness against such augmentations, enabling the acquisition of more discriminative representations and consequently improving model efficacy.
>
> (2) Within the context of UDA for 3D segmentation tasks, an idealized model equipped with perfect structural priors would exhibit invariance to diverse structural perturbations (e.g., heterogeneous scanning methodologies). Previous approaches frequently induce equivariance to structural perturbations, primarily due to the insufficient incorporation of structural priors. We address this limitation by integrating previously neglected structural priors (including Permutation Invariance, Local Consistency, and Geometric Consistency) prior to intermediate feature representation learning. **This strategic integration alleviates equivariance effects, amplifies the encoder's invariance to structural perturbations, and constructs a substantially more robust model.** The resultant performance improvements are empirically substantiated in Table 3 of the main text.
>
> ---
>
> ## Q5:
> The proposed Permutation Invariance and Local Consistency priors all work by altering the processing order of points and then enforcing consistency on the feature representations. It seems to be be conceptually overlapping.
>
> ## A5:
> We would like to further clarify the distinction between these two structural priors.
>
> **Permutation Invariance** leverages different space-filling curves (H-curve, Z-curve) to create diverse scanning sequences. Intuitively, this enables the modeling of long-range dependencies from different perspectives (e.g., top-down versus bottom-up approaches). This structural prior can be conceptualized as consistency across **different viewpoints**, ensuring feature representations remain invariant despite changes in observation sequence.
>
> **Local Consistency** priors utilize statistical partitioning based on **local characteristics**. For instance:
>
> - Cylindrical partitioning recognizes that objects proximal to LiDAR sensors frequently belong to categories like "car" or "motorcycle," while distant points often represent "vegetation" (as validated in the LaserMix).
> - Rectangular partitioning leverages the global Z-axis coordinate as a discriminative metric, where lower Z-values typically correspond to "road" classes, while higher Z-values frequently indicate "building" structures.
>
> By implementing various local partitioning schemes, we generate distinct sub-regions within which we apply sequential ordering. This approach effectively preserves local consistency priors and enhances the model's ability to capture domain-invariant features within contextually similar regions.
>
> Consequently, Permutation Invariance and Local Consistency priors leverage fundamentally distinct structural information, enabling complementary feature representation learning across different organizational hierarchies of the LiDAR data.

---

> > ### Comment · Reviewer_ymd8 · 2025-08-04
> >
> > Thanks for the detailed rebuttal and the reply to my concerns.

---

> > > ### Author Response · Authors · 2025-08-05
> > >
> > > Dear Reviewer  ymd8,
> > >
> > > We sincerely thank the reviewer for their valuable feedback and for acknowledging the detailed rebuttal.
> > >
> > > We truly appreciate your recognition of our efforts in addressing the concerns you raised regarding the computational complexity, the design of target-domain loss, the spatial modeling capability of Mamba, and the role of equivariance and the proposed structural priors. Your comments helped us significantly refine our understanding and communication of the method.
> > >
> > > **If there are any remaining questions or aspects that remain unclear, we would be more than happy to provide further clarification or discussion.** We greatly value your insights, and your additional suggestions would be invaluable for further improving our work.
> > >
> > > We also kindly ask you to consider whether our clarifications sufficiently resolve your original concerns. Should you find the updated explanations satisfactory, we would be truly grateful if you could consider updating your evaluation.
> > >
> > > Thank you again for your time and constructive review.
> > >
> > > Best regards,
> > >
> > > Authors of Submission 16945

---

> > > > ### Comment · Reviewer_ymd8 · 2025-08-05
> > > >
> > > > One additional question: According to the table, the computational load of the BeyondMix++ method during the training stage (76.84 GFLOPs) is approximately 20% higher than that of the DGT-ST baseline (63.68 GFLOPs). This is an expected result, as the new method introduces additional modules and computations. However, the table also shows that the single training iteration time for BeyondMix++ (0.67 seconds) is nearly 30% faster than that of DGT-ST (0.95 seconds). An algorithm with a higher FLOPs should logically have a longer runtime. What is the cause of this?

---

> ### Author Response · Authors · 2025-08-05
>
> Thank you for your question.
>
> Our method introduces additional modules and computations, resulting in a reasonably higher computational load, as reflected by the GFLOPs (76.84 vs. 63.68). However, there appears to be a misunderstanding regarding the time metric reported in Table **R1** of **A1**. Specifically, the "**Time**" column is expressed in **iterations per second (iteration/s)**, rather than **seconds per iteration (s/iteration)**. **A higher iteration/s value indicates a faster training speed.**
>
> In our case, BeyondMix++ achieves 0.67 iteration/s, while DGT-ST achieves 0.95 iteration/s, meaning that BeyondMix++ is approximately 30% slower per iteration, which aligns with the higher GFLOPs you observed.
>
> Please don’t hesitate to let us know if there are any further questions — we would be happy to clarify.
>
> Best regards,
>
> Authors of Submission 16945

---

> > ### Comment · Reviewer_ymd8 · 2025-08-07
> >
> > Okay, although it is indeed slower, the performance improvement is well worth it. It seems the authors haven't uploaded their code—are there any plans to open-source it?

---

> > > ### Author Response · Authors · 2025-08-07
> > >
> > > Dear Reviewer ymd8,
> > >
> > >
> > > Thank you very much for your positive feedback and for acknowledging the trade-off between efficiency and performance.
> > >
> > > We appreciate your interest in our work. Yes, we plan to release the code publicly upon the acceptance of the paper, to facilitate further research and reproducibility.
> > >
> > > Best regards,
> > >
> > > Authors of Submission 16945

---

> > > > ### Author Response · Authors · 2025-08-09
> > > >
> > > > Dear Reviewer ymd8,
> > > >
> > > >
> > > > As the discussion period is approaching its conclusion, I would like to kindly check whether there are any remaining questions or concerns that we have not yet addressed. If there are still any unresolved points, please feel free to let me know at any time, and I will be happy to provide further clarification. If everything is clear, we would be grateful if you could consider updating your evaluation accordingly.
> > > >
> > > > We truly appreciate your time and engagement in this process.
> > > >
> > > >
> > > > Best regards,
> > > >
> > > > Authors of Submission 16945

---

> > > > > ### Comment · Reviewer_ymd8 · 2025-08-09
> > > > >
> > > > > Thanks for the author's detailed response. I will carefully consider the opinions of the other reviewers and the AC to provide an appropriate score.

---

> > > > > > ### Author Response · Authors · 2025-08-09
> > > > > >
> > > > > > Dear Reviewer ymd8,
> > > > > >
> > > > > > We sincerely thank you for the time, effort, and thoughtful attention you have devoted to reviewing our work. Your detailed feedback and engagement during the discussion have been truly valuable in helping us refine our ideas and presentation. We greatly appreciate your consideration and the care you have shown throughout this process.
> > > > > >
> > > > > > Best regards,
> > > > > >
> > > > > > Authors of Submission 16945

---

### Official Review · Reviewer_cNXC · 2025-07-02

**Clarity:** 4
**Significance:** 4
**Originality:** 4
**Rating:** 5
**Confidence:** 5

**Summary:**

This paper proposes BeyondMix, a novel framework that harnesses the capabilities of State Space Models (specifically Mamba) to construct and exploit these structural priors while modeling long-range dependencies that transcend the limited receptive fields of conventional voxel-based approaches.

**Questions:**

1. Figure 1(c) is confusing. What's the difference between the symbol "-" and "x"? Could the author provide a detailed explanation? It would be best for the author to provide the strategies adopted by the specific method.
2. Line 247, BeyondMix++ randomly selects from various existing mixing strategies. Could the author present the result of each mixing strategy? And which strategy best fits the proposed BeyondMix.

**Ethical Concerns:**

["NO or VERY MINOR ethics concerns only"]

**Final Justification:**

Thanks for authors' rebuttal. I haven't found any evidence to lower the rating. I will keep the initial rating.

**Limitations:**

yes

**Paper Formatting Concerns:**

no formatting issues

**Quality:**

4

**Strengths And Weaknesses:**

Strength:
1. This paper identifies three critical yet underexploited structural priors: permutation invariance, local consistency, and geometric consistency.
2. Excellent work. The experiment is complete and sufficient. Writing and drawing are clear and easy to understand.

Weakness:
1. Some multi-modal UDA methods for point cloud segmentation also needs to be discussed in related work.

---

> ### Author Rebuttal · Authors · 2025-07-30
>
> We sincerely appreciate your thorough review and constructive feedback on our manuscript. We are particularly grateful for your recognition of our work as **"excellent"** and your acknowledgment that our experimental validation is **"complete and sufficient"** with **"clear and comprehensible"** illustrations and exposition.
>
> We see that your main concerns are further explanation of the certain ambiguous aspects in our manuscript. All your comments are addressed point by point in the following. **Please feel free to use the discussion period if you have any additional questions.**
>
> ---
>
> ## Q1:
> Some multi-modal UDA methods for point cloud segmentation also needs to be discussed in related work.
>
> ## A1:
>  We appreciate your valuable feedback. **We have supplemented our related work section with multimodality discussions and will incorporate this content into our final manuscript.**
>
> Unsupervised domain adaptation (UDA) for 3D semantic segmentation has garnered increasing academic attention, particularly regarding the exploitation of multimodal information streams. Seminal investigations [1, 2] explored the utilization of auxiliary modalities such as depth—available exclusively during source domain training—to facilitate adaptation of 3D semantic segmentation frameworks. Subsequently, xMUDA [3] established a paradigmatic approach by enforcing cross-modal consistency constraints, thereby enabling bidirectional knowledge transfer between image and point cloud representations to enhance domain generalization capabilities. Further advancements to this paradigm [4] incorporated sophisticated cross-modal fusion mechanisms and contrastive learning objectives to optimize representation alignment across both modalities and domains.
> Contemporary research has extended these foundational approaches through integration of advanced vision architectures and refined fusion methodologies. For instance, [5] incorporates the Segment Anything Model (SAM) to augment 2D modality representations, consequently enhancing 3D segmentation performance through more effective cross-modal knowledge transfer. The authors of [6] propose a sequential fusion-then-distillation framework, which first aligns 2D and 3D feature representations within a shared latent manifold before employing positive distillation techniques to preserve complementary modality-specific information during the adaptation process. Furthermore, [7] introduces an adaptive regularization framework for modality-guided feature fusion, facilitating dynamic and contextually appropriate integration of visual and geometric information under domain distribution shifts.
>
>
>
> [1] Spigan: Privileged adversarial learning from simulation. ICLR, 2019
>
> [2] Dada: Depth-aware domain adaptation in semantic segmentation. ICCV, 2019.
>
> [3] xMUDA: Cross-modal unsupervised domain adaptation for 3D semantic segmentation. CVPR,2020
>
> [4]Cross-modal learning for domain adaptation in 3D semantic segmentation. TPAMI, 2022
>
> [5]Learning to adapt sam for segmenting cross-domain   point clouds. ECCV 2024
>
> [6]Fusion-then-Distillation: Toward Cross-modal Positive Distillation for Domain Adaptive 3D Semantic Segmentation. TCSVT 2025
>
> [7]Exploring Modality Guidance to Enhance VFM-based Feature Fusion for UDA in 3D Semantic Segmentation. CVPR 2025
>
> ---
>
> ## Q2:
> Figure 1(c) is confusing. What's the difference between the symbol "-" and "x"? Could the author provide a detailed explanation? It would be best for the author to provide the strategies adopted by the specific method.
>
> ## A2:
> We apologize for any confusion and will address this issue in the final manuscript.
>
> In our notation, **"-"** denotes structural priors **not previously explored** in the literature, specifically permutation invariance, local consistency, and geometric consistency. The **"x"** symbol indicates structural priors that **cannot be implemented using mix-based methodologies.**
>
> Existing mix-based approaches each leverage a single structural prior:
>
> - **Cosmix** utilizes **Category Semantic Integrity**  by employing semantic category instances as the minimal mixing unit(illustrated in Fig. 1(b), first diagram)
> - **Polarmix** implements **Azimuthal Semantic Consistency** by extracting and rotationally mixing points of identical semantic classes (depicted in Fig. 1(b), second diagram)
> - **Lasermix** enforces **Spatial Region Integrity** by partitioning scenes according to inclination patterns (shown in Fig. 1(b), third diagram)
>
>
> These methodologies are limited by their singular focus on one structural prior and insufficient receptive field capacity.
> Our approach incorporates both mix-compatible priors and those incompatible with mixing techniques. The additional 3 structural priors we adopt are:
>
> - **Permutation Invariance** (Fig. 1(b), fourth diagram): Ensures consistency across different scanning orders representing varied observational perspectives (e.g., top-down versus bottom-up)
> - **Local Consistency**(Fig. 1(b), fifth diagram): Leverages statistical partitioning. For instance:
>    - Cylindrical partitioning recognizes proximity-based semantic patterns (objects near sensors are typically" car"or "motorcycle"  while distant points often represent "vegetation")
>    - Rectangular partitioning utilizes Z-axis coordinates (lower values correspond to" road" while higher values indicate "buildings")
>
> - **Geometric Consistency** (Fig. 1(b), sixth diagram): Maintains consistent geometric representations across varying LiDAR point densities.
>
> Additionally, our method models long-range dependencies, substantially expanding the receptive field and addressing the domain-invariant feature learning limitations in unsupervised domain adaptation caused by restricted receptive fields.
>
> ---
>
> ## Q3:
> Line 247, BeyondMix++ randomly selects from various existing mixing strategies. Could the author present the result of each mixing strategy? And which strategy best fits the proposed BeyondMix.
>
> ## A3:
> As demonstrated in **lines 329-336 of the main text and in Table 7**, we systematically evaluated the performance gains achieved by integrating BeyondMix with previous mixing strategies (CosMix, PolarMix, LaserMix) and their combinations. Our experiments reveal that when combining BeyondMix with individual mixing methods, LaserMix achieves the highest performance (45.4%). We attribute this to **LaserMix's more comprehensive structural priors and effective cross-domain receptive field construction**. When pairing BeyondMix with two mixing methods simultaneously, the CosMix+LaserMix combination yields the best results (46.0%). This can be explained by the operational similarities between PolarMix and LaserMix—**both involve radius-based manipulations relative to the central point, albeit through different mechanisms**—making CosMix+LaserMix a more complementary pairing. The integration of all three strategies achieves optimal performance (46.2%), confirming that our BeyondMix++ successfully leverages the diverse structural priors and strengths of each mixing method to capture a more comprehensive range of domain-invariant features for robust LiDAR semantic segmentation.
>
> **Table 7: Comparison between different mix strategies.**
>
> | #   | CosMix | PolarMix | LaserMix | mIoU (%)        |
> | --- | ------ | -------- | -------- | --------------- |
> | (1) | ✔      |          |          | 45.1 (**−1.1**) |
> | (2) |        | ✔        |          | 44.8 (**−1.4**) |
> | (3) |        |          | ✔        | 45.4 (**−0.8**) |
> | (4) | ✔      | ✔        |          | 45.7 (**−0.5**) |
> | (5) | ✔      |          | ✔        | 46.0 (**−0.2**) |
> | (6) |        | ✔        | ✔        | 45.9 (**−0.3**) |
> | (7) | ✔      | ✔        | ✔        | **46.2**        |

---

> > ### Comment · Reviewer_cNXC · 2025-08-07
> >
> > Thank you for your detailed response. I think it would be better to modify Fig.1(c), because the focus of this paper is to consider and introduce Permutation Invariance, Local Consistency, and Geometric Consistency Priors. Highlighting the differences from other methods. Hence, the appearance of the 3rd row appears redundant. Overall, the author answered my questions.

---

> > > ### Author Response · Authors · 2025-08-07
> > >
> > > Dear Reviewer cNXC,
> > >
> > > Thank you very much for your thoughtful feedback and for recognizing our efforts in addressing your questions.
> > >
> > > We sincerely appreciate your suggestion regarding Fig. 1(c). We agree that the current visualization could be improved to better highlight the key contributions of our work — namely, the modeling of Permutation Invariance, Local Consistency, and Geometric Consistency Priors. In the final version, we will revise the figure accordingly to emphasize the distinctions from prior methods and avoid redundant elements.
> > >
> > > Thank you again for your valuable input！
> > >
> > >
> > >
> > > Best regards,
> > >
> > > Authors of Submission 16945

---

### Note · Authors · 2025-08-12

We thank the AC and reviewers for their time, effort, and constructive feedback during the review and discussion process, and are encouraged by the recognition of our work’s **clear motivation**, **strong empirical results**, **high novelty**, and **clear, well-written presentation**.

**Key Innovations**

Our work is highly innovative in both theory and practice:

1.We formalize three previously underexplored structural priors—Permutation Invariance, Local Consistency, and Geometric Consistency—that effectively capture global context while preserving fine-grained local structures, directly addressing a key challenge in LiDAR-based UDA.

2.We theoretically and empirically demonstrate that long-range dependency modeling via Mamba substantially enhances domain-invariant representation learning in LiDAR segmentation, overcoming the receptive-field limitations inherent in voxel-based methods.

3.By unifying structural priors with long-range modeling, BeyondMix forms a generalizable framework applicable across synthetic-to-real, real-to-real, and cross-sensor adaptation, achieving consistent state-of-the-art results.

**Concerns Addressed**

In the rebuttal and discussion, we resolved all major points:

1.Efficiency – Full runtime, memory, and FLOPs analysis (Table R1) shows inference cost comparable to baselines with modest, justified training overhead.

2.Loss design – Through extensive investigation and ablation experiments, we confirm the necessity of the target-domain loss for mix-based 3D UDA.

3.Spatial modeling – Multiple sequential orderings capture complementary 3D relations beyond voxel limits.

4.Equivariance vs. invariance – Clarified how intermediate equivariance strengthens final prediction invariance.

5.Priors’ distinction – Differentiated conceptual and functional roles of Permutation Invariance vs. Local Consistency.

6.Generalization – Added real-to-real and cross-sensor experiments with consistent SOTA performance.

7.Clarity – Will revise Fig. 1(c) and terminology for precision.

**Commitment**

We will integrate all clarifications, expand related work, and release code upon acceptance.

We believe BeyondMix offers **theoretical depth and practical impact** by **unifying structural priors for global–local context integration**  with **long-range modeling for domain-invariant representation learning**, addressing a fundamental bottleneck in UDA and inspiring future research in 3D domain adaptation.

---

### Decision · Program_Chairs · 2025-09-17

**Decision:**

Accept (poster)

**Comment:**

This paper introduces BeyondMix, a novel framework for unsupervised domain adaptation (UDA) in LiDAR semantic segmentation. BeyondMix leverages State Space Models (specifically Mamba) to model long-range dependencies in point cloud data, utilizing space-filling curves for sequential ordering. The method explicitly formalizes and exploits three structural priors—permutation invariance, local consistency, and geometric consistency—that have been underexploited in existing approaches. Extensive experiments on established benchmarks (SynLiDAR → SemanticKITTI and SemanticPOSS) demonstrate significant improvements over previous state-of-the-art methods.

**Strengths**:
- Clearly identifies key limitations of existing mix-based UDA methods and proposes an innovative solution.
- Explicitly models multiple complementary structural priors and integrates them within a unified framework.
- Employs Mamba for effective long-range context modeling, overcoming receptive field constraints.

**Weaknesses**:

- The additional computational complexity and overhead introduced by feature serializations and Mamba integration are only briefly acknowledged.
- Limited experimental analysis of generalization settings beyond synthetic-to-real adaptation (although partially addressed in the rebuttal).

**Rebuttal**:

The authors presented quantitative analysis of training/inference efficiency, showing comparable computational costs at inference with only modest training overhead. Generalization was substantiated through additional experiments on real-to-real and cross-sensor adaptation, with strong results. Architectural choices (Mamba vs. PointTransformer v3) were discussed both conceptually and empirically, reinforcing the design decisions.

**Suggestions for Improvement**:
- Expand empirical evaluation to encompass additional adaptation scenarios and modalities for broader impact.
- Further document computational/resource overhead and clarify efficiency trade-offs.
- Provide more interpretability on structural priors’ class-wise effects and deeper comparative studies against alternative architectures.